# 📝 STAMP Your Content:
# Proving Dataset Membership via Watermarked Rephrasings

**Saksham Rastogi** [1]  **Pratyush Maini** [2][3]  **Danish Pruthi** [1]

## Abstract

Given how large parts of publicly available text are crawled to pretrain large language models (LLMs), data creators increasingly worry about the inclusion of their proprietary data for model training without attribution or licensing. Their concerns are also shared by benchmark curators whose test-sets might be compromised. In this paper, we present **STAMP**, a framework for detecting dataset membership—i.e., determining the inclusion of a dataset in the pretraining corpora of LLMs. Given an original piece of content, our proposal involves first generating multiple rephrases, each embedding a watermark with a unique secret key. One version is to be released publicly, while others are to be kept private. Subsequently, creators can compare model likelihoods between public and private versions using paired statistical tests to prove membership. We show that our framework can successfully detect contamination across four benchmarks which appear only once in the training data and constitute less than 0.001% of the total tokens, outperforming several contamination detection and dataset inference baselines. We verify that **STAMP** preserves both the semantic meaning and utility of the original data. We apply **STAMP** to two real-world scenarios to confirm the inclusion of paper abstracts and blog articles in the pretraining corpora.

## 1. Introduction

To train large language models, much of the available text from the internet is crawled, allegedly including copyrighted material such as news articles and blogs (Grynbaum & Mac,

2023a;b). Additionally, some evaluation datasets, originally intended for benchmarking model performance, may be compromised—an issue prominently discussed as *test-set contamination* (Magar & Schwartz, 2022; Jacovi et al., 2023; Sainz et al., 2023a). A recent study reveals concerning evidence that pretraining corpora contain several key benchmarks (Elazar et al., 2024), and another demonstrates that impact of test set contamination has been underestimated in many prominent LLM releases (Singh et al., 2024).

On one hand, training language models on copyrighted material might violate legal standards, and on the other, consuming test sets of machine learning benchmarks might offer a false sense of progress. Given the lack of regulations or incentives for model developers to disclose contents of their pretraining corpora (OpenAI, 2023; AI@Meta, 2024; Anthropic, 2024), it is critical to equip content creators with reliable tools to determine whether their content was included as a part of model training. Especially, third party approaches that can democratize detecting dataset membership and enable independent accountability.

Some approaches for detecting dataset membership embed random sequences in text or substitute characters with visually-similar unicodes (Wei et al., 2024). However, such alterations impair machine readability, indexing and retrieval—making them impractical for content creators. More critically for benchmarks, such substitutions can alter tokenization, potentially compromising their utility for evaluation. Other proposals rely on access to a *validation* set that is unseen by the target model and drawn from the same distribution as the original dataset—a requirement hard to meet in practice (Maini et al., 2024). Recently, Oren et al. (2024) suggest comparing canonical ordering of test sets to random permutations, but this strategy assumes large portions of datasets are processed together within a single context window during pretraining. Most closely related to our proposal, Zhang et al. (2024a) use a statistical test to compare model confidence on original test instances and their rephrasings, assuming that the two distributions are identical—an assumption we show does not hold (Table 9).

[1]Indian Institute of Science [2]Carnegie Mellon University [3]DatologyAI. Correspondence to: Saksham Rastogi <iitdsaksham@gmail.com>, Pratyush Maini <pratyush-maini.@cmu.edu>.

*Proceedings of the $42^{nd}$ International Conference on Machine Learning*, Vancouver, Canada. PMLR 267, 2025. Copyright 2025 by the author(s).

We make all our code, data and models available at github.com/codeboy5/STAMP

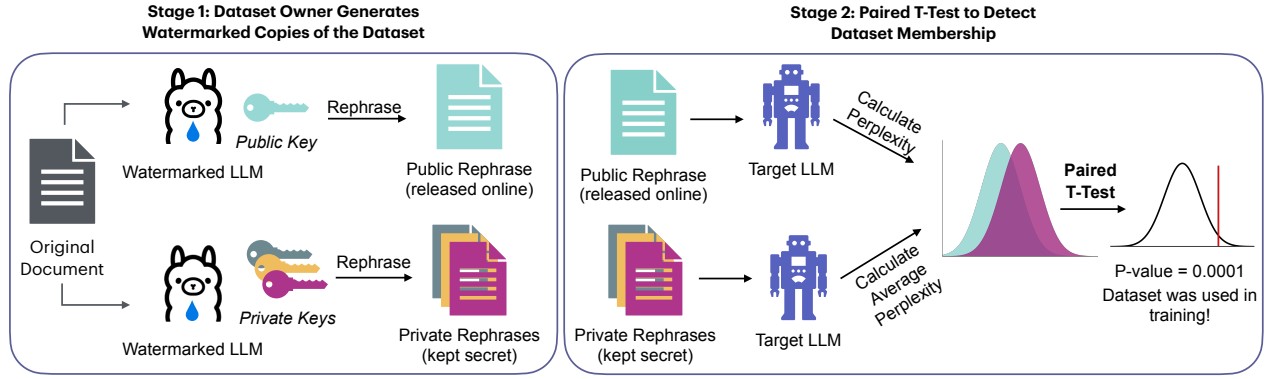

Figure 1: **Overview of STAMP**. **Stage 1:** *Create Watermarked Copies of the Dataset.* We use a watermarked LLM to generate multiple rephrased versions of their original dataset, each uniquely watermarked using a distinct key. The version watermarked with the public key is released publicly on the internet, while other watermarked versions are kept private. **Stage 2:** *Prove Membership using a Paired T-Test.* To detect membership, we compute model perplexities over documents from both the public and private versions. Using these perplexity scores, we perform a paired t-test to detect membership.

In our work, we propose **STAMP** (**S**potting **T**raining **A**rtifacts through water**M**arked **P**airs), a practical approach allowing creators to detect dataset membership through a statistical test with a probabilistic interpretation (Figure 1). Our approach begins by taking the original content and generating multiple rephrased versions. Each rephrased version is watermarked using a distinct key for the hash function used in watermarking. Content creators can then release one of the generations publicly, while keeping the others private. A statistical test then evaluates the model likelihood of generating the public version against the private copies. For models that were trained on the publicly available generations, we expect to observe higher model likelihoods for these generations compared to their private counterparts.

Our work repurposes LLM watermarking to watermark documents that considerably enhance the detection sensitivity of our statistical test. (This is different from watermarking *models* themselves to prevent against model extraction attacks.) Specifically, we leverage the KGW watermarking scheme (Kirchenbauer et al., 2024), which embeds detectable signals by steering generations towards a randomly chosen "green" subset of the vocabulary.

We empirically validate the effectiveness of our approach by continually pretraining the Pythia 1B model (Biderman et al., 2023) on deliberately contaminated pretraining data. We contaminate the pretraining corpus by injecting test examples from four different benchmarks. Even with minimal contamination—that is, each test example appearing only once and each benchmark comprising less than $0.001\%$ of the total training data—our approach significantly outperforms existing methods, achieving statistically significant p-values across all contaminated benchmarks. We also conduct a false positive analysis, wherein we apply our detec-

tion methodology to off-the-shelf pretrained LLMs that have not been exposed to the watermarked benchmarks and find that our test successfully denies their membership. Moreover, our analysis reveals that watermarking substantially enhances detection sensitivity, improving statistical significance by up to three orders of magnitude.

To demonstrate **STAMP**'s effectiveness in detecting inclusion of copyrighted data in pretraining corpora, we present two expository case studies where we apply **STAMP** to detect membership of paper abstracts and blog articles. Our test achieves statistically significant p-values across these real-world scenarios. To further ensure that our framework preserves content quality, we conduct both automatic evaluations using GPT4 (OpenAI, 2023) and a human study, and find that **STAMP** maintains content quality. These results highlight its utility in protecting copyrighted material (for creators), and detecting contamination (for auditors).

## 2. Preliminaries

In this section, we begin by formalizing the problem of detecting membership of a dataset (§2.1) and provide necessary background on watermarks for LLMs (§2.2).

### 2.1. Dataset Membership

The problem of dataset membership (Maini et al., 2021) aims to determine whether a dataset $X$ has been included in the pretraining data $D_{\text{train}}$ of a language model $\theta$. We operate under a *gray-box* setting, where we can compute token probabilities for any sequence $S$ but have no access to the pretraining data or model weights. Formally detecting membership of a dataset can be posed as a hypothesis test with the goal to distinguish between the following hypothesis:

- $H_0$: $\theta$ is independent of $X$ (no membership)

- $H_1$: $\theta$ is dependent on $X$ (membership),

where we treat $\theta$ as a random variable whose randomness arises from the sampling of the pretraining dataset $D_{\text{train}}$ (which may or may not include $X$). Framing membership inference (Shokri et al., 2016) as hypothesis testing provides statistical guarantees on the false detection rate.

**Our focus** is on building statistical tests that can reliably detect dataset membership in language models. We aim to develop methods that make minimal assumptions about the format or nature of data—be it machine learning benchmarks, newsletters, or books.

## 2.2. Watermarks for LLMs

Watermarking techniques for LLMs embed subtle but distinctive patterns within generated text that are imperceptible to humans but algorithmically detectable. For our framework, we utilize the prominent KGW scheme (Kirchenbauer et al., 2024). KGW scheme uses a hash function that takes the context (preceding tokens) and a hash key $h$ to partition the vocabulary $V$ into two disjoint sets at each generation step: a green list $G$ and a red list $R$.

To embed a watermark, the scheme biases the model's next-token probabilities by adding $\delta$ ($\delta > 0$) to the logits of tokens in the green list. Specifically, if $l_k^{(t)}$ denotes the original logit for token $k$ at position $t$, then the modified logits are given by:

$$\hat{l}_k^{(t)} \leftarrow l_k^{(t)} + \delta \mathbb{1}\big[k \in G\big]. \qquad (1)$$

## 3. STAMP: Spotting Training Artifacts through Watermarked Pairs

We introduce **STAMP**, a practical and principled framework that enables content creators to reliably detect whether their content was included in LLM pretraining data. Our approach builds on a key insight: if an LLM consistently prefers documents watermarked with a specific key (e.g., the key used for the publicly available version) over semantically equivalent content with distinct watermarks, then the model must have seen the preferred documents during pretraining. In this section, we detail how **STAMP** leverages this insight to create a robust statistical framework for membership detection. **STAMP** consists of two stages: (1) a process for content creators to release watermarked content (§3.1) and (2) a paired statistical test to detect downstream dataset membership (§3.2).

## 3.1. Watermarking Datasets

The first stage of our approach involves generating multiple watermarked versions of a dataset through rephrasing. For a given dataset $X$, we employ an open-weights instruction-tuned LLM to generate rephrases. For each document $q$ in the original dataset, we create a public version (denoted as $q^{'}$), where the rephrase is watermarked using a designated public key as the *hash key*. Additionally, we generate $m$ private versions (denoted as $q_1^{''}, q_2^{''}, \ldots, q_m^{''}$), where each generation is watermarked using a distinct private key as the hash key. The public version is released online, while the private versions are kept confidential. Crucially, due to the design of our test relying on pairwise comparisons at a document level (§3.2), each document $q$ in a dataset $X$ can use a different set of hash keys. This ensures that introducing watermarking during the rephrasing stage does not alter the token distribution of the dataset $X$ and, importantly, preserves the overall token distribution of the internet data.

**LLM Watermarks as Sampled Markers.** While watermarking is traditionally intended for attributing generated text to a specific LLM, our motivation diverges from this original purpose. First, we leverage LLM watermarking as a mechanism to embed distinct signals into the rephrases through the use of distinct hash keys. The randomness in both our hash key selection and the watermarking process itself enables us to frame the detection problem as hypothesis testing. Under the null hypothesis $H_0$ (no membership), the target model shouldn't favor content watermarked with any particular key. Second, the watermarking process itself introduces subtle perturbations that increase sequence perplexity, which has been empirically shown to enhance memorization during training (Meeus et al., 2024a), further amplifying our ability to detect membership.

## 3.2. Detecting Dataset Membership

To detect membership, we leverage the insight that under the null hypothesis $H_0$ (no membership), the model should not exhibit any systematic preference towards any of the semantically equivalent paraphrases of documents that are watermarked with distinct keys—the public version of the dataset and privately held versions of the dataset. This follows from the randomness inherent in our selection of keys and nature of watermark we employ. We formalize this intuition through a statistical testing framework.

For each document $q$, we compute the perplexity difference $d_i$ between its public version $q_i^{'}$ and private version $q_i^{''}$ that form a pair $(q_i^{'}, q_i^{''})$:

$$d_i = PPL_\theta(q_i^{'}) - PPL_\theta(q_i^{''}). \qquad (2)$$

Prior to applying the paired t-test, we modify the top 5% outliers by clipping their values. This prevents issues where the test can become ineffective due to a few outlier samples. Under the alternative hypothesis $H_1$, we expect these differences to be negative on average, indicating lower perplexity

for public versions. We evaluate this using a one-sided paired t-test statistic:

$$t = \frac{\bar{d}}{s_d/\sqrt{n}}, \tag{3}$$

where $\bar{d}$ and $s_d$ are the mean and standard deviation of the differences across the collection of documents respectively and $n$ is the number of documents (Student, 1908). The one-sided p-value specifically tests for $\bar{d} < 0$, following our alternative hypothesis that exposure during training leads to lower perplexity on public versions. Paired tests provide higher statistical power, enabling detecting membership even with a smaller collection of documents ($n$), as we show in our experiments.

**Multiple Private Keys.** In practice, we empirically observe that models may exhibit inherent biases at an individual document level, occasionally assigning lower perplexity to specific private rephrases ($q''$) independent of membership. To make our detection robust against such biases, we propose using multiple private rephrases, each watermarked with a distinct key. Instead of comparing against a single private version, we test whether public rephrases exhibit lower perplexity compared to the average perplexity across $m$ different private rephrases:

$$d_i = PPL_\theta(q'_i) - \frac{1}{m}\sum_{j=1}^{j=m} PPL_\theta(q''_{i,j}), \tag{4}$$

where $m$ is a hyperparameter known as *private key count*, and $q''_{i,j}$ represents the $j^{th}$ private rephrase of $i^{th}$ document. Through controlled experiments, we analyze the effect of this hyperparameter on statistical strength of our test (§4.4).

# 4. Experiments & Results

To evaluate the ability of **STAMP** for membership detection, we first focus on benchmark contamination—the inclusion of evaluation benchmarks in the pretraining corpora of LLMs. This setting presents unique challenges for membership detection. First, benchmarks must maintain their utility as reliable indicators of progress, which constrains the modifications we can make prior to their release. Second, benchmarks typically contain limited text compared to other content types (e.g., books or newsletters), making detection particularly challenging.

## 4.1. Releasing Watermarked Test Sets

We evaluate our approach using four widely-used benchmarks: TriviaQA (Joshi et al., 2017), ARC-C (Clark et al., 2018), MMLU (Hendrycks et al., 2021), and GSM8K (Cobbe et al., 2021). For each benchmark, we follow our

proposed methodology (§3.1) to generate watermarked public and private paraphrases. We use the instruction tuned Llama3-70B (AI@Meta, 2024) model and a benchmark-agnostic prompt (provided in Appendix K) to generate these rephrased copies. For each benchmark, we randomly select one watermarked version to be the *public* version. Examples of the rephrased test instances are provided in Appendix L.

**Key Distinction.** While rephrasing has been previously explored for detecting contamination (Zhang et al., 2024a), existing approaches typically compare human-written content against their LLM-generated rephrases, overlooking a crucial confounding factor: language models exhibit systematic preferences for LLM-generated text over human-written content (Liu et al., 2023b; Mishra et al., 2023; Laurito et al., 2024). This inherent bias undermines the reliability of statistical approaches that compare human-written content with their LLM rephrasings, as any detected differences might stem from this general preference rather than training exposure. To enable reliable statistical testing, it is crucial to control the data generating process for both versions being compared. We address this by ensuring both our public and private versions are generated through the same process, differing only in their watermarking keys. Given the random selection of keys, we expect no systematic preferences between versions unless one was seen during training.

We empirically validate that human-written content and its LLM-generated rephrasings are easily distinguishable (thus violating the expected IID requirement): a simple bag-of-words classifier obtains AUROC $> 0.8$ on four out of five benchmarks, whereas the classifier performs no better than random chance when distinguishing between rephrasings watermarked with different keys. Detailed analysis and classifier specifications are provided in Appendix C.

## 4.2. Pretraining with Intentional Contamination

**Setup.** To simulate downstream benchmark contamination as it occurs in real-world scenarios and evaluate the effectiveness of our test, we perform continual pretraining on the 1 billion parameter Pythia model (Biderman et al., 2023) using an intentionally contaminated pretraining corpus. The corpus is a combination of OpenWebText (Contributors, 2023) and *public* watermarked version of the four benchmarks, as mentioned in Section 4.1. Each test set accounts for less than **0.001%** of the pretraining corpus, with exact sizes detailed in Table 6 in the appendix. All test sets in our experiments have a duplication rate of 1 (denoting no duplication whatsoever), and the overall pretraining dataset comprises 6.7 billion tokens. Details of the exact training hyperparameters are provided in Appendix E.

**Baselines.** We compare **STAMP** against two recent statistical approaches to detect membership: PaCoST (Zhang et al.,

Table 1: **P-values for detecting *test-set contamination* for different methods.** For LLM DI (Maini et al., 2024), *same* refers to using rephrases of the benchmark questions as *validation* set, while *different* uses an entirely different set of unseen questions from the same benchmark as the *validation* set. **Bold** indicates statistically significant results ($p < 0.05$). Across all the four benchmarks, our approach results in lower p-values compared to other approaches (lower is better).

| | BENCHMARK ($\downarrow$) | | | |
|---|---|---|---|---|
| METHOD | TRIVIAQA | ARC-C | MMLU | GSM8K |
| PACOST (ZHANG ET AL., 2024A) | **1.6E-3** | 0.33 | 0.19 | 0.21 |
| LLM DI (MAINI ET AL., 2024) (*same*) | 0.43 | 0.31 | 0.46 | 0.30 |
| LLM DI (MAINI ET AL., 2024) (*different*) | **0.02** | 0.53 | **0.03** | 0.71 |
| **STAMP** (W/O PAIRED TESTS) | 0.14 | 0.07 | 0.08 | **0.02** |
| **STAMP** (W/O WATERMARKING) | **0.02** | **5.1E-3** | **0.02** | **1.4E-3** |
| **STAMP** | **1.2E-4** | **2.8E-4** | **7.0E-4** | **6.6E-6** |

2024a) and LLM DI (Maini et al., 2024). PaCoST employs a paired t-test that compares model confidence on original and rephrased versions, while LLM DI aggregates multiple membership inference attacks (MIAs) to perform statistical testing. For LLM DI, which requires access to an unseen *validation* set, we evaluate two settings: (1) using private rephrases of the publicly available dataset as the *validation* set, and (2) using an entirely different set of documents from the same distribution as the *validation* set.

Additionally, we also evaluate state-of-the-art MIAs: *PPL* (Yeom et al., 2018), *Zlib* (Carlini et al., 2021), *Min-K* (Shi et al., 2024), *Min-K++* (Zhang et al., 2024c) and *DC-PDD* (Zhang et al., 2024d). Since MIAs rely on a non-trivial detection threshold, we report AUROC scores across two settings: (1) discriminating between public rephrases in training and private rephrases of the same documents, and (2) discriminating between public rephrases in training and unseen documents from the same dataset.

**Main Results.** We compare **STAMP** and baseline methods in Table 1. **STAMP** achieves statistically significantly low p-values (ranging from $10^{-4}$ to $10^{-6}$) across all benchmarks, substantially outperforming existing methods. In contrast, PaCoST detects contamination only on TriviaQA ($p \approx 10^{-3}$), while LLM DI shows significance on just two benchmarks (TriviaQA and MMLU) even with access to validation data of extra test examples.

In our experiments, all MIA methods achieve an AUROC score of $\approx 0.5$ across all benchmarks, indicating performance no better than random guessing. Detailed MIA results and analysis are presented in Table 7.

**False Positive Analysis.** To ensure the robustness of **STAMP** against false positives, we conduct two key experiments. First, we apply our detection methodology to off-the-

Table 2: **False positive analysis.** *Pythia Uncontaminated* denotes the p-values on a pretrained Pythia model that has not been contaminated. *Pythia Contaminated* refers to p-values when testing for membership of *held-out* subsets of datasets on a model contaminated with different subsets of the same datasets. High p-values denote that our approach does not falsely detect membership.

| DATASET | ($\uparrow$) PYTHIA UNCONTAMINATED | ($\uparrow$) PYTHIA CONTAMINATED |
|---|---|---|
| TRIVIAQA | 0.52 | 0.28 |
| ARC-C | 0.31 | 0.56 |
| MMLU | 0.54 | 0.15 |
| GSM8K | 0.38 | 0.47 |
| ABSTRACTS | 0.55 | 0.07 |
| BLOGS | 0.21 | 0.73 |

shelf pretrained LLMs that have not been exposed to the watermarked benchmarks. The results for Pythia 1B, presented in the first column of Table 2, show no false positives. We extend this analysis to models of different sizes and families in Table 8, consistently finding no false positives across all tested models, confirming the robustness of **STAMP** against false positives. Second, we perform a stronger test to evaluate whether **STAMP** detects the membership of the dataset rather than just distributional differences due to different watermarking keys. We create held-out subsets from the same benchmarks and watermark them using the identical public keys used for our contaminated versions. While these held-out sets share the same distribution and watermarking as our training data, they contain entirely different examples. We then apply our detection methodology to test if these *held-out* sets are falsely detected as members in our contaminated Pythia 1B model. The second column of Ta-

Table 3: **Performance of models on the original datasets compared to the watermarked benchmarks.** We evaluate the models using the LM evaluation harness (Gao et al., 2024) with the default settings, comparing performance on original benchmarks against two watermarking approaches: UNICODE substitutions (Wei et al., 2024) and **STAMP**. We find that models obtain comparable performance on **STAMP**-watermarked benchmarks, but crucially, **the relative ranking of LLMs remains unchanged across all benchmarks**, demonstrating the utility of watermarked benchmarks in comparing models.

| DATASET | METRIC | VARIANT | PYTHIA 1B | GEMMA-2 2B | MISTRAL 7B | LLAMA-3 8B | GEMMA-2 9B |
|---|---|---|---|---|---|---|---|
| ARC-C | 0-SHOT | ORIGINAL | 26.1 | 48 | 49.1 | 50.6 | 59.0 |
| | | UNICODE | 21.6 | 37.3 | 39.0 | 41.5 | 49.8 |
| | | **STAMP** | 26.3 | 46.8 | 49.1 | 50.5 | 57.1 |
| MMLU | 5-SHOT | ORIGINAL | 28.1 | 52.9 | 59 | 61.1 | 68.6 |
| | | UNICODE | 28.4 | 45.0 | 51.5 | 55.9 | 63.2 |
| | | **STAMP** | 28.8 | 51.6 | 56 | 61.8 | 68.4 |
| TRIVIAQA | 5-SHOT | ORIGINAL | 12.4 | 52.7 | 67.2 | 68.9 | 70.1 |
| | | UNICODE | 1.1 | 23.6 | 46.0 | 44.3 | 54.8 |
| | | **STAMP** | 11.4 | 51.9 | 65.9 | 66.3 | 68.6 |
| GSM8K | 5-SHOT | ORIGINAL | 1.6 | 25.8 | 34.4 | 51.8 | 65.5 |
| | | UNICODE | 1.5 | 23.1 | 23.3 | 46.7 | 60.8 |
| | | **STAMP** | 2.2 | 27.2 | 37.5 | 54.9 | 65.8 |

ble 2 shows consistently large p-values, indicating **STAMP** successfully refutes membership for these *held-out* sets.

**Performance Without Watermarks Embedded.** To validate our hypothesis that using a watermarked LLM to generate the rephrased copies of the benchmark enhances the statistical strength of our test, we conduct experiments under the same settings as described above (§4.2), but with rephrased copies generated without using a watermarked LLM. The results, presented in Table 1, confirm that incorporating watermarked test sets significantly boosts the statistical power of our test, improving performance by at least two orders of magnitude across all benchmarks.

### 4.3. Utility of Test Sets

Detecting contamination alone is insufficient; the watermarked content should retain the desired properties (for e.g., benchmarks should maintain their utility as reliable indicators of LLM performance). Using the lm-evaluation-harness framework (Gao et al., 2024), we assess five pre-trained LLMs on both original and watermarked benchmarks. Additionally, we measure semantic preservation using the P-SP metric (Wieting et al., 2022).

**Our results**, presented in Table 3, demonstrate that **STAMP**-watermarked variants maintain benchmark utility: LLMs achieve similar absolute performance and the relative rankings of LLMs across all benchmarks are unaffected. In contrast, UNICODE watermark (Wei et al., 2024) significantly

Table 4: **Semantic similarity scores** (P-SP) (Wieting et al., 2022) between original datasets and their watermarked rephrases (higher is better). TRIV-QA and ABS. refers to TriviQA and paper abstracts respectively. For reference: the P-SP value is 0.76 for human-written paraphrases as per a recent study (Krishna et al., 2023).

| | TRIV-QA | ARC-C | MMLU | GSM8K | ABS. |
|---|---|---|---|---|---|
| P-SP | 0.91 | 0.83 | 0.86 | 0.90 | 0.95 |

degrades benchmark utility, with performance drops of up to 20% and does not preserve relative rankings. **STAMP**-watermarked variants also result in high semantic preservation (P-SP scores between 0.83 & 0.91) across all benchmarks. For reference, the average score of human paraphrases is 0.76 as per (Krishna et al., 2023). These results are available in Table 4.

### 4.4. Parameters Affecting the Power of the Test

**Benchmark size.** To analyze the effect of sample size ($n$) on detection power, we evaluate our test on benchmark subsets ranging from 100 to 1000 examples. For each size, we average p-values across 10 runs with different random seeds. Our results, in Figure 2a, demonstrate that our approach works even with just 600 examples, where we consistently achieve low p-values ($\approx 10^{-3}$) across all datasets.

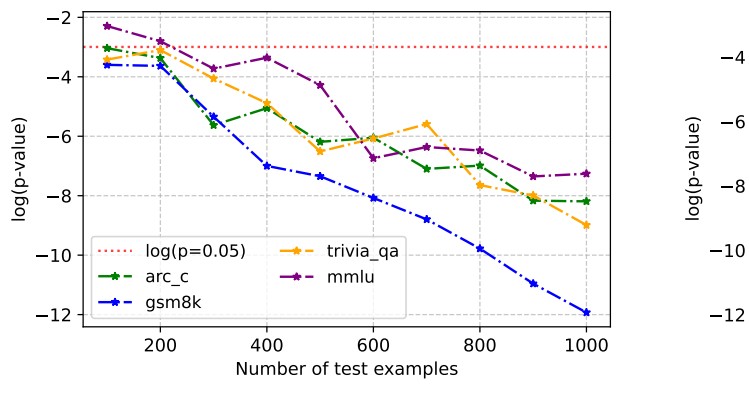

(a) Log p-value vs sample size of benchmark ($n$)

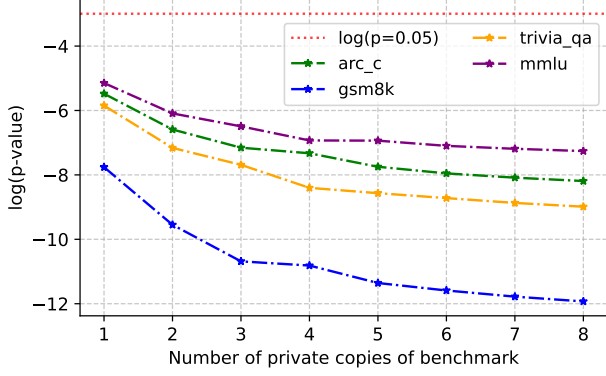

(b) Log p-value vs private key count ($m$)

Figure 2: **Impact of benchmark size ($n$) and private key count ($m$) on STAMP's statistical power.** The dotted red line indicates the standard significance threshold ($p = 0.05$). Lower values indicate stronger statistical evidence of contamination.

**Private key count.** Our proposed test compares the perplexity of the public version against the average perplexity of $m$ private versions (Equation 4). Here we analyze how this hyperparameter ($m$) affects the statistical power of our test. As shown in Figure 2b, increasing the number of private keys strengthens detection up to a threshold of 5 keys, beyond which we see negligible improvement.

**Size of Pretraining Corpora.** We analyze our test's effectiveness for different scales of pretraining data by combining contaminated benchmarks with varying amounts of Open-WebText data (Contributors, 2023). We note that while the strength decreases with corpus size, the *rate of decline* diminishes substantially beyond 4 billion tokens, with minimal drop in detection strength between 4 and 6 billion tokens (Figure 3). Notably, these results are obtained with a modest 1B-parameter model; given that larger models exhibit stronger memorization (Carlini et al., 2019), we believe that STAMP will detect membership for larger models.

## 5. Real World Case Studies

To demonstrate STAMP's effectiveness in detecting *unlicensed* use of copyrighted data in model training, we present two expository case studies. Specifically, we apply STAMP to detect membership of (1) abstracts from EMNLP 2024 proceedings (emn, 2001) and (2) articles from the AI Snake Oil newsletter (Narayanan & Kapoor, 2023).

**Paper Abstracts.** We sample 500 papers from EMNLP 2024 proceedings (emn, 2001) and generate watermarked rephrasings of their abstracts. Additionally, we generate watermarked rephrasings for another set of 500 abstracts, which we use as a *held-out* validation set for our experiments. The prompt templates used for rephrasing and examples of watermarked abstracts are provided in Appendix K

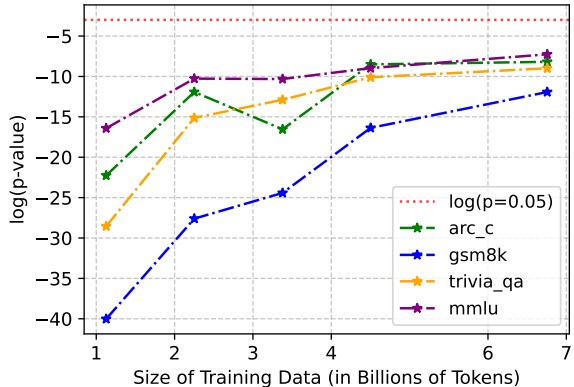

Figure 3: **Log p-value vs pretraining corpus size.** We observe that the *rate of decline* diminishes as we increase the corpus size, with negligible drop between 4B and 6B.

and Appendix L , respectively.

To evaluate whether the semantic content of abstracts is preserved, we use the P-SP metric (Wieting et al., 2022), where watermarked abstracts achieve a high score of 0.95, indicating that the semantic content is largely preserved. To further evaluate the acceptability of watermarked abstracts, we conduct both an automated evaluation (using GPT-4) and a small-scale human study involving original authors. In both evaluations, the original abstract and its watermarked rephrasing is compared, classifying the rephrased content into one of five options: *preferred*, *acceptable*, acceptable with *minor revisions*, or *major revisions*, and lastly *unacceptable*. Further details are available in Appendix I.

For 1000 watermarked abstracts, 99% were rated by GPT-4 as either *preferred* or *acceptable*. In a preliminary human study, we ask authors to review rephrasings of their own

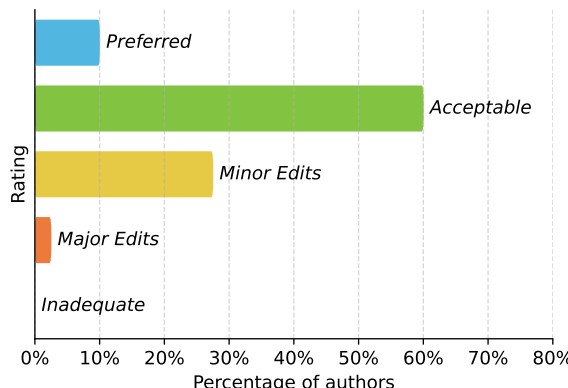

Figure 4: **Author evaluation of watermarked abstracts.** Out of the 40 total abstracts, for 4 of them the watermarked version was *preferred*, for another 24 it was found to be *acceptable*, 11 required *minor edits* with just 1 requiring *major edits*, suggesting that watermarking preserve quality.

abstracts. Our results in Figure 4 show that out of the 40 watermarked abstracts evaluated, authors find 24 to be acceptable as is, indicate 11 could use minor edits, and 4 prefer the rephrased version over their self-written abstracts, with just 1 abstract requiring major edits. Details of the evaluation are provided in Appendix I.

**Blog Posts from AI Newsletter.** We collect 56 posts from the popular AI Snake Oil newsletter (Narayanan & Kapoor, 2023), and use 44 for pretraining and hold 12 for validation. To demonstrate how **STAMP** could handle longer-form content, we adapt it to rephrase at the paragraph level, treating each paragraph as an independent datapoint for our test. Note, while each paragraph serves as a datapoint for our test, the blog posts are included in the pretraining corpora at the document level, following standard pretraining practices (detailed in Appendix E). We present the prompt used to rephrase in Appendix K

We evaluate **STAMP**'s ability to detect dataset membership by performing continual pretraining on the Pythia 1B model using a training corpus composed of watermarked paper abstracts ($\approx$ 105K tokens), watermarked blog posts ($\approx$ 95K tokens), and a subset of OpenWebText ($\approx$ 3.3B tokens). Additionally, to verify that **STAMP** can detect dataset membership for distinct datasets watermarked with the same key, we apply a consistent watermarking key when generating the public versions of both datasets.

**Results.** Our results in Table 5 demonstrate that **STAMP** effectively detects dataset membership for both paper abstracts and blog posts, achieving statistically significant p-values. To compare, we evaluate LLM DI under different choices of validation set: first, using private rephrases of

Table 5: **Case studies.** We report p-values of different approaches for detecting dataset membership (lower is better). LLM DI (same) uses the private rephrasing of the same documents, while LLM DI (different) uses different documents from a held out set from the same distribution.

| METHOD ($\downarrow$) | PAPER ABSTRACTS | BLOG ARTICLES |
|---|---|---|
| LLM DI (SAME) | 0.15 | 0.44 |
| LLM DI (DIFFERENT) | **0.05** | 0.58 |
| **STAMP** (W/O PAIRED TESTS) | **0.01** | 0.07 |
| **STAMP** | **2.7E-12** | **2.4E-3** |

the same documents and second, using a different held-out set of documents watermarked with the same public key. While LLM DI can detect membership for paper abstracts, it fails to do so for blog posts. Further, membership inference attacks exhibit near-random performance (Table 7). To verify the robustness of **STAMP** against false positives, we evaluate it under the two settings discussed earlier (§4.2). Our results in Table 2 confirm that **STAMP** does not result in any false positives, reinforcing its reliability.

## 6. Related Work

We discuss the most closely related work below, focusing on statistical approaches for detecting dataset membership, test-set contamination and the use of watermarks for detecting membership of datasets. A more comprehensive review of related literature is provided in Appendix G.

**Dataset Membership.** A recent hypothesis-testing approach embeds random sequences in text or substitutes characters with visually-similar unicodes (Wei et al., 2024). Similarly, Meeus et al. (2024a) propose inserting "copyright traps" into documents to enhance document-level membership inference. These methods then test the model's preference for these inserted sequences or substitutions. However, such alterations impair machine readability, making them impractical for content creators. Another recent proposal (Maini et al., 2024) selectively combines membership inference attacks (MIAs) that provide positive signals for a given distribution and aggregates them to perform a statistical test on a dataset. Their method assumes access to a validation set drawn from the same distribution as the target dataset and unseen by the model—a difficult requirement to satisfy.

A recent position paper (Zhang et al., 2024b) argues that methods attempting to estimate FPR by collecting non-members a posteriori are statistically unsound, a position that aligns with our analysis of PaCoST (Zhang et al.,

2024a). We believe **STAMP** aligns with the criteria for a sound membership proof presented in the paper. Specifically, we use private members sampled from the same distribution as the publicly released version $x$. Since our private members are semantically equivalent to the public member, any causal effects of publishing $x$ would similarly affect the private members, ensuring the statistical validity of our approach.

**Test Set Contamination.** While our focus is detecting membership of any arbitrary collection of documents, some recent *statistical approaches* have focused on detecting test set contamination. A recent work (Oren et al., 2024) proposes a permutation test based on the canonical ordering in a benchmark but relies on the strong assumption of metadata contamination (canonical ordering of the dataset). Another recent proposal (Zhang et al., 2024a) compares the model confidence on test instances and their rephrased counterparts. However, as discussed earlier (§4.1), LLMs may favor their own outputs, and this is an oft-overlooked confounder. Additionally, there have been a few approaches based on prompting models to reproduce near-exact test examples (Sainz et al., 2023b; Golchin & Surdeanu, 2024). However, the heuristic-y nature of these approaches prevents them from providing statistical evidence of contamination.

**Watermarking for Dataset Membership.** A few recent approaches have explored using LLM watermarks for membership detection. Waterfall (Lau et al., 2024) proposes a watermarking scheme for protecting IP of text and further demonstrates how to detect unauthorized fine-tuning of LLMs on proprietary text data. Specifically, to detect membership of a text, their approach prompts the target model with a prefix and detects the embedded watermarking in the generated new tokens to test for membership. While effective in certain scenarios, their approach requires a higher level of memorization and has only been demonstrated in fine-tuning settings with multiple epochs. Additionally, their method is not applicable to domains like benchmarks where each sample is only a few tokens long. These limitations may restrict its practical utility for detecting membership of a dataset in the pretraining corpora of an LLM.

Another recent contemporaneous study (Sander et al., 2025) proposes a similar approach where watermarks are embedded in benchmarks by reformulating the original questions with a watermarked LLM. While employing a similar setup, their detection approach differs substantially from ours. Their method relies on detecting overfitting of the contaminated model on token-level watermarking biases to prove contamination, whereas our approach compares perplexity differences between the publicly released benchmarks and private versions watermarked with different keys.

## 7. Conclusion & Future Directions

In this work, we presented **STAMP**, a statistical framework for detecting dataset membership, which can reliably be used by content creators to watermark their content, while preserving the utility, or the meaning, of the original content. We demonstrated **STAMP**'s effectiveness in detecting test-set contamination through comprehensive experiments. Our ablation studies systematically analyzed how detection strength varies with dataset size, the number of private versions, and pretraining corpus size. We validated the real-world applicability of our approach through two case studies: detecting paper abstracts and blog posts in pretraining data.

There are several **important limitations** of our work: first, watermarks must be embedded before the content is released online, making it inapplicable to already published content. We believe this is a fundamental limitation shared by existing statistical methods, as they require knowledge of the data-generating process to construct a valid null distribution. Second, our method requires access to token probabilities from the model (gray box access). Third, while our human study showed that majority of authors found the rephrasings to be acceptable, rephrasing could introduce errors in the content. However, we believe this will be less of a concern moving forward as general model capabilities, including paraphrasing quality, continue to improve. Finally, due to computational constraints, we evaluated our approach using continual pretraining rather than training models from scratch. While our results demonstrate effectiveness in this setting outperforming baselines, future work could validate these findings using models that are trained from scratch.

Future work could explore the optimal watermarking strength for different data distributions (and use cases) to balance a (plausible) trade-off between detectability and quality of watermarked content. Future work could also validate, or extend, our approach to other domains, such as code, speech, images or videos.

## Acknowledgments

We thank the reviewers for their feedback. We sincerely thank all participants in our evaluation study for their valuable time. This work was supported in part by the AI2050 program at Schmidt Sciences (Grant G-24-66186). Additionally, DP is grateful to Adobe Inc., Pratiksha Trust and the National Payments Corporation of India (NPCI) for generously supporting his group's research.

## Impact Statement

Our work studies the problem of detecting unauthorized usage of data for model training. In the current landscape, where model developers are reluctant to share details about

their pretraining corpora, we believe our proposal holds potential to considerably increase transparency in model training. Our tool could be beneficial to content creators seeking to protect their work from unauthorized use.

Additionally, our work has implications for the broader AI ecosystem. By detecting test-set contamination, our approach can help researchers obtain more accurate estimates of model capabilities and track AI progress.

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

Contributors, O. Opencompass: A universal evaluation platform for foundation models, 2023.

Das, D., Zhang, J., and Tramèr, F. Blind baselines beat membership inference attacks for foundation models, 2024. URL https://arxiv.org/abs/2406.16201.

Duan, M., Suri, A., Mireshghallah, N., Min, S., Shi, W., Zettlemoyer, L., Tsvetkov, Y., Choi, Y., Evans, D., and Hajishirzi, H. Do membership inference attacks work on large language models? *ArXiv preprint*, abs/2402.07841, 2024. URL https://arxiv.org/abs/2402.07841.

Elazar, Y., Bhagia, A., Magnusson, I., Ravichander, A., Schwenk, D., Suhr, A., Walsh, E. P., Groeneveld, D., Soldaini, L., Singh, S., Hajishirzi, H., Smith, N. A., and Dodge, J. What's in my big data? In *The Twelfth International Conference on Learning Representations, ICLR 2024, Vienna, Austria, May 7-11, 2024*. OpenReview.net, 2024. URL https://openreview.net/forum?id=RvfPnOkPV4.

Gao, L., Tow, J., Abbasi, B., Biderman, S., Black, S., DiPofi, A., Foster, C., Golding, L., Hsu, J., Le Noac'h, A., Li, H., McDonell, K., Muennighoff, N., Ociepa, C., Phang, J., Reynolds, L., Schoelkopf, H., Skowron, A., Sutawika, L., Tang, E., Thite, A., Wang, B., Wang, K., and Zou, A. A framework for few-shot language model evaluation, 2024. URL https://zenodo.org/records/12608602.

Golchin, S. and Surdeanu, M. Time travel in llms: Tracing data contamination in large language models. In *The Twelfth International Conference on Learning Representations, ICLR 2024, Vienna, Austria, May 7-11, 2024*. OpenReview.net, 2024. URL https://openreview.net/forum?id=2Rwq6c3tvr.

Grynbaum, M. M. and Mac, R. The times sues openai and microsoft over a.i. use of copyrighted work https://www.nytimes.com/2023/12/27/business/media/new-york-times-open-ai-microsoft-lawsuit.html, 2023a. URL https://www.nytimes.com/2023/12/27/business/media/new-york-times-open-ai-microsoft-lawsuit.html.

Grynbaum, M. M. and Mac, R. Sarah silverman and authors sue openai and meta over copyright infringement, 2023b. URL https://www.nytimes.com/2023/07/10/arts/sarah-silverman-lawsuit-openai-meta.html.

Hendrycks, D., Burns, C., Basart, S., Zou, A., Mazeika, M., Song, D., and Steinhardt, J. Measuring massive multitask language understanding. In *9th International Conference on Learning Representations, ICLR 2021, Virtual Event, Austria, May 3-7, 2021*. OpenReview.net, 2021. URL https://openreview.net/forum?id=d7KB jmI3GmQ.

Jacovi, A., Caciularu, A., Goldman, O., and Goldberg, Y. Stop uploading test data in plain text: Practical strategies for mitigating data contamination by evaluation benchmarks. In Bouamor, H., Pino, J., and Bali, K. (eds.), *Proceedings of the 2023 Conference on Empirical Methods in Natural Language Processing*, pp. 5075–5084, Singapore, 2023. Association for Computational Linguistics. doi: 10.18653/v1/2023.emnlp-main.308. URL https://aclanthology.org/2023.emnlp-m ain.308.

Joshi, M., Choi, E., Weld, D., and Zettlemoyer, L. TriviaQA: A large scale distantly supervised challenge dataset for reading comprehension. In Barzilay, R. and Kan, M.-Y. (eds.), *Proceedings of the 55th Annual Meeting of the Association for Computational Linguistics (Volume 1: Long Papers)*, pp. 1601–1611, Vancouver, Canada, 2017. Association for Computational Linguistics. doi: 10.18653/v1/P17-1147. URL https://aclantho logy.org/P17-1147.

Kirchenbauer, J., Geiping, J., Wen, Y., Katz, J., Miers, I., and Goldstein, T. A watermark for large language models. In Krause, A., Brunskill, E., Cho, K., Engelhardt, B., Sabato, S., and Scarlett, J. (eds.), *International Conference on Machine Learning, ICML 2023, 23-29 July 2023, Honolulu, Hawaii, USA*, volume 202 of *Proceedings of Machine Learning Research*, pp. 17061–17084. PMLR, 2023. URL https://proceedings.mlr.pres s/v202/kirchenbauer23a.html.

Kirchenbauer, J., Geiping, J., Wen, Y., Shu, M., Saifullah, K., Kong, K., Fernando, K., Saha, A., Goldblum, M., and Goldstein, T. On the reliability of watermarks for large language models. In *The Twelfth International Conference on Learning Representations, ICLR 2024, Vienna, Austria, May 7-11, 2024*. OpenReview.net, 2024. URL https://openreview.net/forum?id= DEJIDCmWOz.

Krishna, K., Song, Y., Karpinska, M., Wieting, J., and Iyyer, M. Paraphrasing evades detectors of ai-generated text, but retrieval is an effective defense. In Oh, A., Naumann, T., Globerson, A., Saenko, K., Hardt, M., and Levine, S. (eds.), *Advances in Neural Information Processing Systems 36: Annual Conference on Neural Information Processing Systems 2023, NeurIPS 2023, New Orleans, LA, USA, December 10 - 16, 2023*, 2023. URL http: //papers.nips.cc/paper_files/paper/2 023/hash/575c450013d0e99e4b0ecf82bd1 afaa4-Abstract-Conference.html.

Lau, G. K. R., Niu, X., Dao, H., Chen, J., Foo, C.-S., and Low, B. K. H. Waterfall: Framework for robust and scalable text watermarking and provenance for llms, 2024. URL https://arxiv.org/abs/2407.04411.

Laurito, W., Davis, B., Grietzer, P., Gavenčiak, T., Böhm, A., and Kulveit, J. Ai ai bias: Large language models favor their own generated content. *ArXiv preprint*, abs/2407.12856, 2024. URL https://arxiv.org/ abs/2407.12856.

Liu, Y., Hu, H., Chen, X., Zhang, X., and Sun, L. Watermarking text data on large language models for dataset copyright. *ArXiv preprint*, abs/2305.13257, 2023a. URL https://arxiv.org/abs/2305.13257.

Liu, Y., Iter, D., Xu, Y., Wang, S., Xu, R., and Zhu, C. G-eval: NLG evaluation using gpt-4 with better human alignment. In Bouamor, H., Pino, J., and Bali, K. (eds.), *Proceedings of the 2023 Conference on Empirical Methods in Natural Language Processing*, pp. 2511–2522, Singapore, 2023b. Association for Computational Linguistics. doi: 10.18653/v1/2023.emnlp-main.153. URL https://aclanthology.org/2023.emnlp-m ain.153.

Loshchilov, I. and Hutter, F. Decoupled weight decay regularization. In *7th International Conference on Learning Representations, ICLR 2019, New Orleans, LA, USA, May 6-9, 2019*. OpenReview.net, 2019. URL https: //openreview.net/forum?id=Bkg6RiCqY7.

Magar, I. and Schwartz, R. Data contamination: From memorization to exploitation. In Muresan, S., Nakov, P., and Villavicencio, A. (eds.), *Proceedings of the 60th Annual Meeting of the Association for Computational Linguistics (Volume 2: Short Papers)*, pp. 157–165, Dublin, Ireland, 2022. Association for Computational Linguistics. doi: 10.18653/v1/2022.acl-short.18. URL https: //aclanthology.org/2022.acl-short.18.

Maini, P., Yaghini, M., and Papernot, N. Dataset inference: Ownership resolution in machine learning. In *9th International Conference on Learning Representations, ICLR 2021, Virtual Event, Austria, May 3-7, 2021*. OpenReview.net, 2021. URL https://openreview.net /forum?id=hvdKKV2yt7T.

Maini, P., Jia, H., Papernot, N., and Dziedzic, A. LLM dataset inference: Did you train on my dataset? In Globersons, A., Mackey, L., Belgrave, D., Fan, A., Paquet, U., Tomczak, J. M., and Zhang, C. (eds.), *Advances in Neural Information Processing Systems 38: Annual*

*Conference on Neural Information Processing Systems 2024, NeurIPS 2024, Vancouver, BC, Canada, December 10 - 15, 2024*, 2024. URL http://papers.nips.cc/paper_files/paper/2024/hash/e01519b47118e2f51aa643151350c905-Abstract-Conference.html.

Meeus, M., Shilov, I., Faysse, M., and de Montjoye, Y. Copyright traps for large language models. In *Forty-first International Conference on Machine Learning, ICML 2024, Vienna, Austria, July 21-27, 2024*. OpenReview.net, 2024a. URL https://openreview.net/forum?id=LDq1JPdc55.

Meeus, M., Shilov, I., Jain, S., Faysse, M., Rei, M., and de Montjoye, Y.-A. Sok: Membership inference attacks on llms are rushing nowhere (and how to fix it). 2024b. URL https://api.semanticscholar.org/CorpusID:270737841.

Mishra, A., Rahman, S., Kim, H. J., Mitra, K., and Hruschka, E. R. Characterizing large language models as rationalizers of knowledge-intensive tasks. *ArXiv preprint*, abs/2311.05085, 2023. URL https://arxiv.org/abs/2311.05085.

Narayanan, A. and Kapoor, S. Ai snake oil., 2023. URL https://www.aisnakeoil.com/. Newsletter.

NewYorkTimes. The times sues openai and microsoft over a.i. use of copyrighted work. https://www.nytimes.com/2023/12/27/business/media/new-york-times-open-ai-microsoft-lawsuit.html, 2023.

OpenAI. Gpt-4 technical report, 2023. URL https://arxiv.org/abs/2303.08774.

Oren, Y., Meister, N., Chatterji, N. S., Ladhak, F., and Hashimoto, T. Proving test set contamination in black-box language models. In *The Twelfth International Conference on Learning Representations, ICLR 2024, Vienna, Austria, May 7-11, 2024*. OpenReview.net, 2024. URL https://openreview.net/forum?id=KS8mIvetg2.

Sainz, O., Campos, J., García-Ferrero, I., Etxaniz, J., de Lacalle, O. L., and Agirre, E. NLP evaluation in trouble: On the need to measure LLM data contamination for each benchmark. In Bouamor, H., Pino, J., and Bali, K. (eds.), *Findings of the Association for Computational Linguistics: EMNLP 2023*, pp. 10776–10787, Singapore, 2023a. Association for Computational Linguistics. doi: 10.18653/v1/2023.findings-emnlp.722. URL https://aclanthology.org/2023.findings-emnlp.722.

Sainz, O., Campos, J. A., García-Ferreroa, I., and andEneko Agirre, J. E. Did chatgpt cheat on your test? https://hitz-zentroa.github.io/lm-contamination/blog/, 2023b.

Sander, T., Fernandez, P., Durmus, A., Douze, M., and Furon, T. Watermarking makes language models radioactive. In Globersons, A., Mackey, L., Belgrave, D., Fan, A., Paquet, U., Tomczak, J. M., and Zhang, C. (eds.), *Advances in Neural Information Processing Systems 38: Annual Conference on Neural Information Processing Systems 2024, NeurIPS 2024, Vancouver, BC, Canada, December 10 - 15, 2024*, 2024. URL http://papers.nips.cc/paper_files/paper/2024/hash/2567c95fd41459a98a73ba893775d22a-Abstract-Conference.html.

Sander, T., Fernandez, P., Mahloujifar, S., Durmus, A., and Guo, C. Detecting benchmark contamination through watermarking. *ArXiv preprint*, abs/2502.17259, 2025. URL https://arxiv.org/abs/2502.17259.

Shi, W., Ajith, A., Xia, M., Huang, Y., Liu, D., Blevins, T., Chen, D., and Zettlemoyer, L. Detecting pretraining data from large language models. In *The Twelfth International Conference on Learning Representations, ICLR 2024, Vienna, Austria, May 7-11, 2024*. OpenReview.net, 2024. URL https://openreview.net/forum?id=zWqr3MQuNs.

Shokri, R., Stronati, M., Song, C., and Shmatikov, V. Membership inference attacks against machine learning models, 2016. URL https://arxiv.org/abs/1610.05820.

Singh, A. K., Kocyigit, M. Y., Poulton, A., Esiobu, D., Lomeli, M., Szilvasy, G., and Hupkes, D. Evaluation data contamination in llms: how do we measure it and (when) does it matter? *ArXiv preprint*, abs/2411.03923, 2024. URL https://arxiv.org/abs/2411.03923.

Student. The probable error of a mean. *Biometrika*, pp. 1–25, 1908.

Tang, L., Laban, P., and Durrett, G. MiniCheck: Efficient fact-checking of LLMs on grounding documents. In Al-Onaizan, Y., Bansal, M., and Chen, Y.-N. (eds.), *Proceedings of the 2024 Conference on Empirical Methods in Natural Language Processing*, pp. 8818–8847, Miami, Florida, USA, 2024. Association for Computational Linguistics. doi: 10.18653/v1/2024.emnlp-main.499. URL https://aclanthology.org/2024.emnlp-main.499/.

Thai, K., Karpinska, M., Krishna, K., Ray, B., Inghilleri, M., Wieting, J., and Iyyer, M. Exploring document-level literary machine translation with parallel paragraphs

from world literature. In Goldberg, Y., Kozareva, Z., and Zhang, Y. (eds.), *Proceedings of the 2022 Conference on Empirical Methods in Natural Language Processing*, pp. 9882–9902, Abu Dhabi, United Arab Emirates, 2022. Association for Computational Linguistics. doi: 10.186 53/v1/2022.emnlp-main.672. URL https://aclant hology.org/2022.emnlp-main.672.

Wei, J. T.-Z., Wang, R. Y., and Jia, R. Proving membership in llm pretraining data via data watermarks. *ArXiv preprint*, abs/2402.10892, 2024. URL https://arxiv.org/abs/2402.10892.

Wieting, J., Gimpel, K., Neubig, G., and Berg-kirkpatrick, T. Paraphrastic representations at scale. In Che, W. and Shutova, E. (eds.), *Proceedings of the 2022 Conference on Empirical Methods in Natural Language Processing: System Demonstrations*, pp. 379–388, Abu Dhabi, UAE, 2022. Association for Computational Linguistics. doi: 10.18653/v1/2022.emnlp-demos.38. URL https://aclanthology.org/2022.emnlp-demos.38.

Xu, Z., Yuan, S., Chen, L., and Yang, D. "a good pun is its own reword": Can large language models understand puns? In Al-Onaizan, Y., Bansal, M., and Chen, Y.-N. (eds.), *Proceedings of the 2024 Conference on Empirical Methods in Natural Language Processing*, pp. 11766–11782, Miami, Florida, USA, 2024. Association for Computational Linguistics. doi: 10.18653/v1/2024.e mnlp-main.657. URL https://aclanthology.o rg/2024.emnlp-main.657/.

Yeom, S., Giacomelli, I., Fredrikson, M., and Jha, S. Privacy risk in machine learning: Analyzing the connection to overfitting. In *2018 IEEE 31st computer security foundations symposium (CSF)*, pp. 268–282. IEEE, 2018.

Zebaze, A., Sagot, B., and Bawden, R. Tree of problems: Improving structured problem solving with compositionality, 2024. URL https://arxiv.org/abs/2410 .06634.

Zhang, H., Lin, Y., and Wan, X. Pacost: Paired confidence significance testing for benchmark contamination detection in large language models, 2024a. URL https://arxiv.org/abs/2406.18326.

Zhang, J., Das, D., Kamath, G., and Tramèr, F. Membership inference attacks cannot prove that a model was trained on your data, 2024b. URL https://arxiv.org/abs/2409.19798.

Zhang, J., Sun, J., Yeats, E., Ouyang, Y., Kuo, M., Zhang, J., Yang, H. F., and Li, H. Min-k%++: Improved baseline for detecting pre-training data from large language models. *ArXiv preprint*, abs/2404.02936, 2024c. URL https://arxiv.org/abs/2404.02936.

Zhang, W., Zhang, R., Guo, J., de Rijke, M., Fan, Y., and Cheng, X. Pretraining data detection for large language models: A divergence-based calibration method, 2024d. URL https://arxiv.org/abs/2409.14781.

## A. Additional Results

Table 6: Size of evaluation benchmark used in the intentional contamination experiment (§4.4). Each benchmark is subsampled to 1,000 examples, with each injected benchmark making up less than 0.001% of the entire pretraining corpus, which consists of 6.7 billion tokens. Each benchmark is injected exactly once into the corpus without any duplication.

| BENCHMARK | SIZE (TOKENS) | % PRETRAINING DATA |
|---|---|---|
| TRIVIAQA | 34609 | 5.1E-4 |
| ARC-C | 36863 | 5.5E-4 |
| MMLU | 42548 | 6.3E-4 |
| GSM8K | 61132 | 9.0E-4 |

Table 7: **Comparison of Membership Inference Attacks (MIA) performance across different datasets.** We report AUC scores for three MIA methods under two settings: *Same Documents:* public rephrases in training vs private rephrases of the same documents, and *Different Documents:* public rephrases in training vs different unseen documents from the same dataset. AUROC score of $\approx 0.5$ indicates performance no better than random guessing.

| DATASET | Same Documents. (↑) | | | | | Different Documents. (↑) | | | | |
|---|---|---|---|---|---|---|---|---|---|---|
| | PPL | ZLIB | MIN-K | MIN-K++ | DC-PDD | PPL | ZLIB | MIN-K | MIN-K++ | DC-PDD |
| TRIVIAQA | 0.50 | 0.50 | 0.50 | 0.50 | 0.52 | 0.46 | 0.57 | 0.48 | 0.44 | 0.58 |
| ARC-C | 0.50 | 0.50 | 0.50 | 0.49 | 0.51 | 0.49 | 0.50 | 0.48 | 0.45 | 0.52 |
| MMLU | 0.48 | 0.49 | 0.48 | 0.49 | 0.52 | 0.43 | 0.48 | 0.44 | 0.45 | 0.52 |
| GSM8K | 0.50 | 0.50 | 0.50 | 0.50 | 0.52 | 0.47 | 0.47 | 0.48 | 0.48 | 0.52 |
| PAPER ABSTRACTS | 0.48 | 0.49 | 0.48 | 0.46 | 0.53 | 0.41 | 0.46 | 0.42 | 0.40 | 0.55 |
| BLOG ARTICLES | 0.50 | 0.51 | 0.51 | 0.49 | 0.50 | 0.49 | 0.51 | 0.48 | 0.46 | 0.51 |

Table 8: **False positive analysis on off-the-shelf LLMs.** We apply **STAMP** on LLMs that have not seen the datasets and report the p-values. Our results (high p-values) show that our method is robust against false positives.

| DATASET (↑) | PYTHIA 1B | GEMMA-2 2B | MISTRAL 7B | LLAMA-3 8B | GEMMA-2 9B |
|---|---|---|---|---|---|
| TRIVIAQA | 0.52 | 0.91 | 0.94 | 0.65 | 0.91 |
| ARC-C | 0.31 | 0.25 | 0.12 | 0.26 | 0.37 |
| MMLU | 0.54 | 0.41 | 0.29 | 0.24 | 0.43 |
| GSM8K | 0.38 | 0.16 | 0.26 | 0.71 | 0.37 |
| PAPER ABSTRACTS | 0.55 | 0.74 | 0.83 | 0.63 | 0.89 |
| BLOG ARTICLES | 0.21 | 0.72 | 0.74 | 0.88 | 0.12 |

### A.1. Detecting Partial Contamination

In practice, benchmarks may be partially contaminated, where only a subset of test examples appears in the pretraining corpora. Understanding the impact of partial contamination is critical because benchmark owners cannot identify which specific test examples have been leaked. This study complements our earlier analysis in Section 4.4, by focusing on the sensitivity of our approach under varying proportions ($\alpha$) of contaminated examples within a fixed benchmark size ($n$).

Our results in Figure 5 highlight that as $\alpha$ increases the detection strength improves, with p-values dropping below $10^{-3}$ when majority of the benchmark is contaminated. We also observe that **STAMP** reliably detects contaminated even when only 40% of the test examples are contaminated. Our findings confirm that **STAMP** successfully identifies contamination with high statistical significance, even in scenarios of partial contamination.

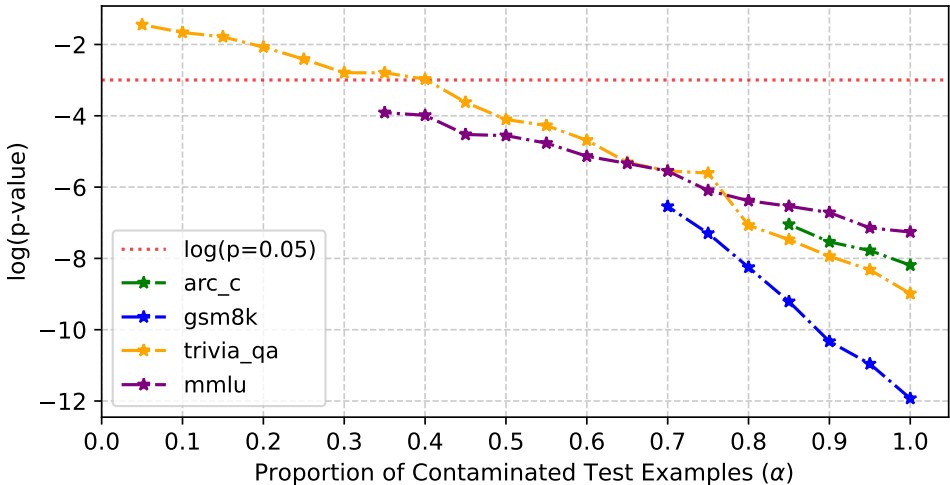

Figure 5: **Log p-value vs proportion of benchmark that is contaminated**. We plot the log p-value against the proportion of test examples that are leaked to analyze the sensitivity of our test to detect contaminated in scenarios where the benchmark is only partially contaminated (lower is better).

## B. P-SP Metric

To validate semantic preservation in our watermarking process, we employ P-SP (Wieting et al., 2022), a state-of-the-art semantic similarity model. P-SP uses embedding averaging trained on a large corpus of filtered paraphrase data, and has been shown to effectively distinguish between true paraphrases and unrelated text. As evidenced by Krishna et al. (2023), P-SP assigns an average score of 0.76 to human-created paraphrases in the PAR3 dataset (Thai et al., 2022), while random paragraph pairs from the same book score only 0.09. Table 4 reports the average P-SP scores between original benchmarks and their watermarked versions across 9 random *hash keys*. Our watermarked versions achieve high P-SP scores (0.83-0.95) across all benchmarks, substantially exceeding the average score for human paraphrases, indicating strong semantic preservation.

## C. Bag-of-Words Classifier

We train a random forest classifier on the *bag-of-words* feature representations for the datasets. The classifier is trained on 80% of the member and non-member sets, with evaluation performed on the remaining 20%. Results are aggregated over a 5-fold cross-validation. The detailed results are presented in Table 9.

Table 9: AUROC using *bag-of-words* features to distinguish between different versions of datasets. The first column shows AUROC for distinguishing original datasets from their rephrased versions, where high values ($> 0.8$) indicate clear distributional differences. The second column shows AUROC for distinguishing between *public* and *private* watermarked versions, where values near $0.5$ indicate distributional similarity.

| DATASET | ORIGINAL VS REPHRASED | PUBLIC VS PRIVATE |
|---|---|---|
| TRIVIAQA | 0.66 | 0.51 |
| ARC-C | 0.83 | 0.52 |
| MMLU | 0.83 | 0.53 |
| GSM8K | 0.84 | 0.57 |
| PAPER ABSTRACTS | 0.86 | 0.57 |

## D. Perplexity

Perplexity (*PPL*) measures how well a language model predicts a given text sequence $S$, with lower values indicating better prediction. For an auto-regressive language model $\theta$ and text sequence $S$, tokenized as a sequence of N tokens $\{s_1, \ldots, s_N\}$, perplexity is computed as the exponent of the loss. Formally:

$$PPL_\theta(S) = \exp\left(\mathcal{L}_\theta(S)\right) \tag{5}$$

Where the loss $\mathcal{L}_\theta$ is defined as:

$$\mathcal{L}_\theta(S) = -\frac{1}{N}\sum_{i=1}^{N} \log\left(\mathcal{P}_\theta(s_i|s_{<i})\right) \tag{6}$$

Here $\mathcal{P}_\theta(s_i|s_{<i})$ denotes the predicted probability for token $s_i$ by the language model $\theta$ given the context of previous tokens $\{s_1, \ldots, s_{i-1}\}$.

## E. Pretraining Details

We continually pretrain Pythia 1B on intentionally contaminated OpenWebText Test case instances from the benchmark were randomly inserted between documents from OpenWebText. We trained for 1 epoch of 46000 steps with an effective batch size of 144 sequences and sequence length of 1024 tokens. We used the AdamW optimizer (Loshchilov & Hutter, 2019) with a learning rate of $10^{-4}$, $(\beta_1, \beta_2) = (0.9, 0.999)$ and no weight decay.

## F. Watermark for Large Language Models

In work, we use the prominent KGW (Kirchenbauer et al., 2023) watermarking scheme. KGW scheme uses a hash function that takes the context (preceding tokens) and a hash key $h$ to partition the vocabulary $V$ into two disjoint sets at each generation step: a green list $G$ and a red list $R$. Formally, for a language model $\mathcal{M}$ with vocabulary $V$, and a prefix comprising tokens $\mathbf{w}_1, \mathbf{w}_2, \ldots, \mathbf{w}_n$, the scheme involves first computing the logits $\mathcal{M}(\mathbf{w}_1 \ldots, \mathbf{w}_n) = (l_1, \ldots, l_{|V|})$ of the language model that would ordinarily be used to predict the subsequent token. As per a hyper-parameter $k$, the last $k$ tokens, $\mathbf{w}_{n-k+1}$ to $\mathbf{w}_n$, are then fed to a pseudo-random function $F$ to partition $V$ into a green list $G$ and a red list $R$ such that $|G| + |R| = |V|$. Finally, the logits corresponding to the tokens in the green list, $G$, are boosted by $\delta$ $(\delta > 0)$. Specifically, in our work we set $k = 1$ and $\delta = 1.0$ as the chose hyperparameters. The watermark can then be detected through a one-proportion z-test on the fraction of green tokens in the generated text.

## G. Related Works

Our works relates to a large literature of work on membership inference (§G.2), dataset membership (§G.3) and test-set contamination detection (§G.4) in large language models.

### G.1. Watermarking for Dataset Membership

Liu et al. (2023a) propose TextMarker, a backdoor-based membership inference technique for protection of *classification datasets*. Their approach watermarks each original sample $(x, y)$ by inserting specific triggers (character or word-level substitutions) into $x$, creating a backdoored sample $x_t$. They then assign an altered target label $y_t$ $(y_t \neq y)$ to this modified input. Membership detection is performed by testing whether a model $f$ produces the watermarked label with high probability: $\Pr(f(x_t) = y_t)$. While effective, TextMarker is specifically designed for classification datasets and it is not trivial to extend it to other kinds of benchmarks or the broader problem of dataset membership detection, limiting its real-world applicability.

Recent work has proposed detecting watermark signals in suspect model outputs to determine dataset membership. Waterfall (Lau et al., 2024) enables creators to watermark their text using a robust watermarking scheme that leverages an LLM for rephrasing. They further demonstrate that watermarks in fine-tuning data, persist in downstream LLM outputs, allowing for membership detection. While both approaches use LLM-based rephrasing, Waterfall relies on detecting watermarks directly

in model generations, whereas our method uses multiple watermarked versions to detect perplexity divergences. While Waterfall is effective for text watermarking, it has limited utility for pretraining membership detection. It relies on strong overfitting and presumes that the model is trained on datasets over multiple epochs. Additionally, their approach is not applicable to short text segments commonly found in benchmarks. Another recent contemporaneous study (Sander et al., 2025) proposes a similar approach where watermarks are embedded in benchmarks by reformulating the original questions with a watermarked LLM. While employing a similar setup, their detection approach differs substantially from ours. Their method relies on detecting model overfitting on the green tokens in the watermarked benchmark to prove contamination, whereas our approach compares perplexity differences between the publicly released benchmarks and private versions watermarked with different keys.

### G.2. Membership Inference

Membership inference, initially proposed by Shokri et al. (2016), is a long-standing problem in machine learning: given a data point and a machine learning model, determine whether that data point was used to train the model. MIAs for LLMs are broadly based on applying pre-defined thresholds to membership scores that are typically based on *loss-based* metrics. We briefly describe the specific membership scores proposed by different MIAs that we employ in our experiments.

- PERPLEXITY: Proposed by Yeom et al. (2018), this MIA uses loss (perplexity in the context of LMs) as the scoring metric. However, this approach suffers from high false positives as it tends to classify naturally predictable sequences as members of the training set.

- ZLIB ENTROPY (Carlini et al., 2021) computes a score by taking the ratio between the model's perplexity and the zlib compression size of the text. Lower ratios indicate potential membership in the training data.

- MIN-K% (Shi et al., 2024) computes the score by averaging the probabilities of the $k\%$ least likely tokens in a sequence. By focusing on the least likely tokens, it aims to solve the false positive problem with perplexity.

- MIN-K%++ (Zhang et al., 2024c) compares the probability of the target token with the expected probability of all tokens within the vocabulary. It is based on the insight that each training token will tend to have higher probability relative to many other candidate tokens in the vocabulary.

- DC-PPD (Zhang et al., 2024d) computes the divergence between the token probability distribution and the token frequency distribution for detection.

### G.3. Detecting Dataset Membership

Detecting dataset membership addresses the challenge of detecting whether a given dataset was used by LLM developers in pretraining. Unlike membership inference attacks (MIAs), which focus on identifying whether individual sequences were included in a model's training data, dataset membership concerns verifying the inclusion of a collection of documents.

Wei et al. (2024) propose a hypothesis-testing approach to detect membership by inserting random sequences or Unicode character substitutions as data watermarks. This method works by testing the model's preference for the inserted data watermarks against other random data watermarks. First, their proposed watermarks can impact machine readability, affecting search engine indexing and retrieval-augmented generation (RAG) pipelines. More critically, unicode substitutions can significantly alter tokenization processes, potentially compromising the utility of evaluation benchmarks. Although these limitations may be manageable for some creators, Our approach offers an alternative that better preserves content quality while maintaining detection capability. Another recent proposal (Maini et al., 2024) is to selectively combining MIAs that provide positive signal for a given distribution, and aggregating them to perform a statistical test on a given dataset. Their method assumes access to a *validation* set drawn from the same distribution as the target dataset and unseen by the model–a requirement that can be challenging to satisfy in many practical scenarios.

Meeus et al. (2024a) propose inserting "copyright traps" into documents to enhance document-level membership inference for smaller models that lack natural memorization. Liu et al. (2023a) introduce a backdoor-based dataset inference approach. However, these methods rely on heuristics and do not provide the false positive guarantees that hypothesis-testing-based approaches offer.

Recent studies (Maini et al., 2024; Duan et al., 2024; Das et al., 2024; Meeus et al., 2024b) suggest that detecting sequence level membership in LLMs trained on trillions of tokens in a single epoch is likely infeasible. These studies also highlight

the limited efficacy of MIAs for LLMs, showing that such approaches barely outperform random guessing. Moreover, the apparent success of MIAs in certain scenarios can often be attributed to distributional differences between the *member* and *non-member* sets used in evaluations, rather than their ability to reliably infer true membership.

### G.4. Test Set Contamination Detection

There have been a few recent third-party approaches that are focused on detecting test-set contamination in LLMs. Heuristic prompting-based methods (Sainz et al., 2023b; Golchin & Surdeanu, 2024) attempt to detect contamination by prompting models to reproduce exact or near-exact test examples. Reproducing verbatim examples requires a high level of memorization which typically requires a high duplication of test examples (Carlini et al., 2021) and strong memorization capabilities typically absent in smaller models (Meeus et al., 2024a). The heuristic nature of these approaches prevents them from providing a statistical evidence of contamination.

Statistical approaches to detect contamination are limited. Oren et al. (2024) build on the principle that in absence of data contamination, all orderings of an *exchangeable* test set should be equally likely. Their work relies on the strong assumption of metadata contamination (canonical ordering of the dataset)–a presumption that can often be violated. Another recent proposal (Zhang et al., 2024a) uses a statistical test to compare model confidence on original test instances and their rephrased counterparts. However, as discussed earlier, their null hypothesis can be invalid due to LLMs' inherent bias towards machine-generated content.

## H. Radioactivity of Watermarks

Sander et al. (2024) proposed methods to detect when watermarked texts are used as fine-tuning data for an LLM. Their approach is based on the insight that training on watermarked texts leaves detectable traces of the watermark signal in the resulting model due to token-level overfitting. In a recent contemporaneous study, Sander et al. (2025) extended this approach to detect benchmark contamination. Specifically, they propose watermarking benchmarks before release and later detecting traces left by the watermarked benchmarks through a statistical test. Since the statistical test relies on token-level overfitting, their approach requires duplication and stronger watermarks, which introduce more distortion into the rephrasings. Additionally, the tokenizer-dependent nature of detecting watermarks limits the applicability of their approach, as the rephrasing model and contaminated model need to share the same tokenizer.

Given the requirement that the rephrasing and contaminated LLM should share the same tokenizer, we conduct additional controlled experiments comparing the approaches. We rephrase with Llama-3.1-8B Instruct (AI@Meta, 2024) with top-p sampling with $p = 0.7$ and temperature $= 0.5$, matching the sampling parameters used in the original study. For watermarking, we use KGW (Kirchenbauer et al., 2024) scheme, with context window of size 2, split ratio ($\gamma$) of $0.5$ & and boosting value ($\delta$) of 2. We create a contaminated corpus of 2 billion tokens following our methodology in Section 4.2, but with a duplication count of 4, meaning each benchmark sample is inserted four times in the pretraining corpora.

We compare **STAMP** with Sander et al. (2025) for detecting benchmark contamination in Table 10. With a moderate watermarking strength ($\delta = 2.0$) and repetition count of 4, the radioactivity based approach fails to detect contamination of watermarked test examples, while **STAMP** achieves significantly low p-values. These results align with the original paper's findings, which indicated that their method requires around 16 repetitions to achieve low p-values.

Table 10: **P-values for detecting *test-set contamination*.** We compare our proposed **STAMP** approach with detection based on radioactivity (Sander et al., 2025). Rows marked **0** denote vanishingly small p-values. Across both the benchmarks, **STAMP** consistently achieves lower p-values (lower is better).

| | BENCHMARK ($\downarrow$) | |
| :---: | :---: | :---: |
| METHOD | ARC-C | MMLU |
| RADIOACTIVITY (SANDER ET AL., 2025) | 0.61 | 0.13 |
| **STAMP** | **0** | **0** |

# I. Case Study: Detecting Research Paper Abstracts in Pretraining Data

To demonstrate the broader applicability of STAMP for detecting dataset membership across different forms of content, we explore its effectiveness in detecting membership of abstracts of papers from EMNLP '24 proceedings (emn, 2001). We evaluate both the preservation of academic writing quality in watermarked abstracts and the effectiveness of STAMP in detecting their inclusion in training data.

**Experimental Setup.** We sample 500 papers from EMNLP 2024 proceedings and create watermarked versions of their abstracts following our methodology from Section 3. The prompt template and examples of rephrased abstracts are presented in Appendix K and Appendix L.2 respectively. To evaluate detection capability, we perform controlled experiments on the Pythia 1B model (Biderman et al., 2023) through continual pretraining. The pretraining corpora consists of a mixture of the public watermarked versions of these abstracts and a subset of OpenWebText (approximately 3 billion tokens). The abstracts comprise approximately 100K tokens, representing just 0.003% of the pretraining corpus.

**Results.** Table 11 demonstrates STAMP's effectiveness in detecting dataset membership. Our approach achieves a near-zero p-value ($\approx 10^{-12}$), indicating strong statistical evidence of membership. For comparison, LLM DI (Maini et al., 2024) achieves a p-value of 0.05 with access to a *validation* set of unseen abstracts from the same conference and is unable to detect membership using the privately held counterparts of the same abstracts included in the pretraining data as the *validation* set. In Table 7 we evaluate state-of-the-art MIAs and finding that they perform no better than random chance (AUROC $\approx 0.5$). Our findings corroborate with recent studies (Duan et al., 2024; Maini et al., 2024; Das et al., 2024) that highlight the failure of sequence level MIAs on LLMs.

**Quality Evaluation.** To evaluate the quality of watermarked abstracts, we use GPT-4 (OpenAI, 2023) as a judge following the prompt template in Figure I. Each abstract was classified into one of five quality tiers. Our analysis shows that 82.7% of the watermarked abstracts were rated as *preferred* and 16.3% as *acceptable* indicating that 99% maintain high academic quality. Only 1% required *minor revisions*, with none requiring *major revisions* or deemed *inadequate*.

Since LLMs often exhibit systematic preferences for LLM-generated text over human-written content (Liu et al., 2023b; Mishra et al., 2023; Laurito et al., 2024), we additionally conduct a human study involving the original authors. We asked 40 authors to rate watermarked versions of their own abstracts using the same quality tiers. The human evaluation strongly corroborates our automatic assessment, with most watermarked versions being *preferred* or *acceptable*: 4 authors *preferred* the watermarked version, 24 authors rated the watermarked abstracts as *acceptable*, 11 indicated the text required *minor revisions* and just 1 indicating that their rephrased abstract requires major edits.

Additionally, we measure semantic preservation using the P-SP metric (Wieting et al., 2022), finding an average score of **0.95** between original and watermarked abstracts, demonstrating strong semantic similarity.

Table 11: **Comparison of different approaches for detecting membership of paper abstracts. Bold** indicates statistically significant results ($p < 0.05$). Our approach results in lower p-values compared to other approaches (lower is better).

| METHOD | P-VALUE ($\downarrow$) |
|---|---|
| LLM DI (MAINI ET AL., 2024) (1) | 0.15 |
| LLM DI (MAINI ET AL., 2024) (2) | **0.05** |
| STAMP (W/O PAIRED TESTS) | **0.01** |
| STAMP | **2.7E-12** |

Prompt Template to Evaluate Quality of the Rephrased Abstracts using GPT4

```
You will be given an original abstract and its rephrased version.  Your task
is to evaluate the quality of abstract rewrites for ML research paper based
on:

1.  Meaning Preservation
2.  Clarity
3.  Technical Accuracy

Evaluate the rewritten abstract and assign one of these ratings:
- Preferred:  The rewrite improves upon the original in terms of clarity and
readability while maintaining full technical accuracy.
- Acceptable:  The rewrite matches the original in quality and could serve
as a direct replacement without requiring changes.
- Minor Revisions:  The rewrite is promising but requires minor edits to
reach the original's quality.
- Major Revisions:  The rewrite has significant issues with meaning
preservation, clarity, or technical accuracy and requires major edits.
- Inadequate:  The rewrite fails to convey the original research effectively
due to critical flaws in meaning, clarity, or technical accuracy.

Here are the abstracts:

Original Abstract:  {original_abstract}
Rephrased Abstract:  {watermarked_abstract}

Provide a short explanation of your rating, followed by your final rating
in the format:
Final Rating:  {rating}
```

## J. Case Study: Detecting ML Blog Posts in Pretraining Data

The inclusion of copyrighted material in LLM training data has emerged as a significant concern, leading to legal disputes, such as the lawsuit between New York Times and OpenAI (NewYorkTimes, 2023), among others. Through a case study, we demonstrate how **STAMP** can help creators detect potential unauthorized use of their content in model training. Specifically, we use **STAMP** to detect the membership of the popular AI Snake Oil newsletter (Narayanan & Kapoor, 2023).

**Experimental Setup.**   We collect 56 blogs from the newsletter, creating watermarked versions of each newsletter using, the prompt template is presented in Figure K. We randomly select a subset of 44 blogs that we include in pretraining corpora and keep the remaining 12 blogs as a *validation* set that is unseen by the model. To evaluate detection capability, we perform controlled experiments on the Pythia 1B model (Biderman et al., 2023) through continual pretraining. The pretraining corpora consists of a mixture of the public watermarked versions of these abstracts and a subset of OpenWebText (approximately 3 billion tokens). The abstracts comprise approximately 94K tokens, representing just 0.003% of the pretraining corpus.

**Results.**   Table 11 demonstrates **STAMP**'s effectiveness in detecting dataset membership for the blog articles. LLM DI is unable to detect membership under the two different choices of validation set: (1) with the private rephrases of the same 44 blog posts as the validation set, and (2) with the version of the *held out* set of 12 blog posts that is watermarking using the public key. In Table 7 we evaluate state-of-the-art MIAs and finding that they perform no better than random chance (AUROC ≈ 0.5). Our findings corroborate with recent studies (Duan et al., 2024; Maini et al., 2024; Das et al., 2024) that highlight the failure of sequence level MIAs on LLMs.

Table 12: **Comparison of different approaches for detecting membership of AI Snake Oil. Bold** indicates statistically significant results ($p < 0.05$). Our approach results in lower p-values compared to other approaches (lower is better).

| METHOD | P-VALUE ($\downarrow$) |
|---|---|
| LLM DI (MAINI ET AL., 2024) (1) | 0.44 |
| LLM DI (MAINI ET AL., 2024) (2) | 0.58 |
| **STAMP** (W/O PAIRED TESTS) | 0.07 |
| **STAMP** | **2.4E-3** |

## K. Prompt Templates for Rephrasing

In this section, we outline the prompts used with LLaMA-3 70B (AI@Meta, 2024) to generate watermarked versions of the documents used in our experiments.

---

**Prompt Template for Rephrasing Benchmarks**

```
Rephrase the question given below.  Ensure you keep all details present in
the original, without omitting anything or adding any extra information not
present in the original question.

Question:  What is the main energy source for deep ocean currents that move
large volumes of water around the planet?

Your response should end with "Rephrased Question:  [rephrased question]"
```

---

**Prompt Template for Rephrasing Abstracts**

```
Rephrase the abstract of a ML research paper given below following these
strict guidelines:

PRESERVE:
- All technical details and findings
- Original tone of the abstract

AVOID:
- Adding interpretive language not present in the original abstract
- Removing any details
- Changing meaning or emphasis

Abstract:  {original_abstract}

Your response should end with "Rephrased Abstract:  {rephrased_abstract}"
```

> Prompt Template for Rephrasing Blogs
>
> ```
> Rephrase the below paragraph from an AI newsletter while maintaining
> coherent flow between paragraphs.  Here are your instructions:
>
> 1.  I will provide the previous paragraph (marked as CONTEXT) and the
> current paragraph to rephrase (marked as TARGET).
> 2.Your task is to:
> - Rephrase the TARGET paragraph so it flows naturally from the previous
> paragraph (CONTEXT)
> - Keep the same tone and emphasis as the original paragraph
> -Preserve the technical details present in the original paragraph
> - Do not add any extra information not present in the original paragraph
> - Avoid making sentences wordier or adding interpretive language
>
> 3.  Format your response as:  REPHRASED PARAGRAPH: [your rephrased version]
>
> Context:  {context}
> Paragraph:  {paragraph}
> ```

## L. Watermarked Examples

### L.1. Watermarked Test Sets

#### L.1.1. TRIVIAQA

> **Original Question:**  Which enduring cartoon character was created by Bob
> Clampett for the 1938 cartoon Porky's Hare Hunt?
>
> **Rephrased Question:**  Which long-lasting cartoon character was originally
> created by Bob Clampett for the 1938 cartoon titled 'Porky's Hare Hunt'?

> **Original Question:**  Which US state lends its name to a baked pudding, made
> with ice cream, sponge and meringue?
>
> **Rephrased Question:**  Which US state is the namesake of a baked pudding that
> consists of sponge, meringue, and ice cream?

#### L.1.2. ARC CHALLENGE

> **Original Question:**  Company X makes 100 custom buses each year.  Company
> Y makes 10,000 of one type of bus each year.  Which of the following is
> the most likely reason a customer would buy a bus from company X instead
> of company Y?
>
> **Rephrased Question:**  What is the most probable reason a customer would
> choose to purchase a bus from Company X, which produces 100 custom buses
> annually, over Company Y, which manufactures 10,000 buses of a single type
> each year?

**Original Question:** Sugars are necessary for human cell function. Which of the following are human cells not capable of doing?

**Rephrased Question:** Given that sugars are necessary for human cell function, what is it that human cells are unable to do?

### L.1.3. MMLU

**Original Question:** Noradrenaline is the neurotransmitter between which of the two structures below?

**Rephrased Question:** Between which two structures listed below does noradrenaline act as the neurotransmitter?

**Original Question:** On which surfaces of the teeth is dental plaque most likely to accumulate in the mouth of a patient with poor oral hygiene?

**Rephrased Question:** In a patient with poor oral hygiene, on which surfaces of the teeth is dental plaque accumulation most probable in the mouth?

### L.1.4. GSM8K

**Original Question:** Darrell and Allen's ages are in the ratio of 7:11. If their total age now is 162, calculate Allen's age 10 years from now.

**Rephrased Question:** If the current ages of Darrell and Allen are in a 7:11 ratio and their combined age at present is 162, what will be Allen's age a decade from now.

**Original Question:** Indras has 6 letters in her name. Her sister's name has 4 more letters than half of the letters in Indras' name. How many letters are in Indras and her sister's names?

**Rephrased Question:** If Indras' name contains 6 letters, and her sister's name has a number of letters that is 4 more than half the number of letters in Indras' name, then how many letters are there in Indras' and her sister's names combined?

## L.2. Watermarked Paper Abstracts

Tree of Problems: Improving structured problem solving with compositionality (Zebaze et al., 2024)

**Original Abstract:**

Large Language Models (LLMs) have demonstrated remarkable performance across multipletasks through in-context learning. For complex reasoning tasks that require step-by-step thinking, Chain-of-Thought (CoT) prompting has given impressive results, especially when combined with self-consistency. Nonetheless, some tasks remain particularly difficult for LLMs to solve. Tree of Thoughts (ToT) and Graph of Thoughts (GoT) emerged as alternatives, dividing the complex problem into paths of subproblems. In this paper, we propose Tree of Problems (ToP), a simpler version of ToT, which we hypothesise can work better for complex tasks that can be divided into identical subtasks. Our empirical results show that our approach outperforms ToT and GoT, and in addition per forms better than CoT on complex reasoning tasks. All code for this paper will be made available.

**Watermarked Abstract:**

Large Language Models (LLMs) have achieved notable performance in multiple tasks using in-context learning. For complex reasoning tasks requiring step-by-step thinking, Chain-of-Thought (CoT) prompting, especially when combined with self-consistency, has yielded impressive results. However, certain tasks continue to pose significant challenges for LLMs. Tree of Thoughts (ToT) and Graph of Thoughts (GoT) have emerged as alternatives by breaking down complex problems into subproblem paths. This paper introduces Tree of Problems (ToP), a simplified ToT version that we hypothesize will be more effective for complex tasks comprising identical subtasks. Our empirical findings indicate that ToP outperforms both ToT and GoT, and also surpasses CoT on complex reasoning tasks. The code used in this paper will be made available.

"A good pun is its own reword": Can Large Language Models Understand Puns? (Xu et al., 2024)

**Original Abstract:**

Puns play a vital role in academic research due to their distinct structure and clear definition, which aid in the comprehensive analysis of linguistic humor. However, the understanding of puns in large language models (LLMs) has not been thoroughly examined, limiting their use in creative writing and humor creation. In this paper, we leverage three popular tasks, i.e., pun recognition, explanation and generation to systematically evaluate the capabilities of LLMs in pun understanding. In addition to adopting the automated evaluation metrics from prior research, we introduce new evaluation methods and metrics that are better suited to the in-context learning paradigm of LLMs. These new metrics offer a more rigorous assessment of an LLM's ability to understand puns and align more closely with human cognition than previous metrics. Our findings reveal the \lazy pun generation" pattern and identify the primary challenges LLMs encounter in understanding puns.

**Watermarked Abstract:**

Puns, due to their distinct structure and clear definition, play a vital role in academic research on linguistic humor analysis. Despite their importance, the understanding of puns in large language models (LLMs) remains understudied, which hinders their application in creative writing and humor generation. This paper employs three tasks – pun recognition, explanation, and generation – to systematically evaluate LLMs' capabilities in pun understanding. Building upon prior research's automated evaluation metrics, we propose new evaluation methods and metrics tailored to the in-context learning paradigm of LLMs. These novel metrics provide a more rigorous assessment of an LLM's pun understanding ability and better align with human cognition. Our results expose the "lazy pun generation" pattern and pinpoint the primary challenges LLMs face in understanding puns.

MiniCheck: Efficient Fact-Checking of LLMs on Grounding Documents (Tang et al., 2024)

**Original Abstract:**

Recognizing if LLM output can be grounded in evidence is central to
many tasks in NLP: retrieval-augmented generation, summarization,
document-grounded dialogue, and more.  Current approaches to this kind
of fact-checking are based on verifying each piece of a model generation
against potential evidence using an LLM. However, this process can be very
computationally expensive, requiring many calls to a model to check a single
response.  In this work, we show how to build small fact-checking models
that have GPT-4-level performance but for 400x lower cost.  We do this by
constructing synthetic training data with GPT-4, which involves creating
realistic yet challenging instances of factual errors via a structured
generation procedure.  Training on this data teaches models to check each
fact in the claim and recognize synthesis of information across sentences.
For evaluation, we unify datasets from recent work on fact-checking and
grounding LLM generations into a new benchmark, LLM-AggreFact.  Our best
system MiniCheck-FT5 (770M parameters) outperforms all systems of comparable
size and reaches GPT-4 accuracy.  We release LLM-AggreFact, code for data
synthesis, and models.

**Watermarked Abstract:**

Verifying the evidential grounding of LLM outputs is crucial for various
NLP tasks, including retrieval-augmented generation, summarization, and
document-grounded dialogue.  Existing fact-checking approaches typically
involve verifying each component of model generation against potential
evidence using an LLM, which can be computationally expensive and require
multiple model calls for a single response.  This work demonstrates the
construction of compact fact-checking models that achieve GPT-4-level
performance at a 400x lower cost.  We accomplish this by generating
synthetic training data using GPT-4 through a structured procedure that
creates realistic yet challenging instances of factual errors.  Models
trained on this data learn to verify facts within claims and recognize
information synthesis across sentences.  We establish a unified benchmark,
LLM-AggreFact, by consolidating datasets from recent fact-checking and
LLM grounding research.  Our top-performing system, MiniCheck-FT5 (770M
parameters), outperforms comparable-sized systems and matches GPT-4's
accuracy.  We make LLM-AggreFact, the data synthesis code, and the models
publicly available.

