# OpenReview forum: "STAMP Your Content: Proving Dataset Membership via Watermarked Rephrasings"
_ICML.cc/2025/Conference — ICML 2025 poster_

### Official Review · Reviewer_oVrP · 2025-03-03

**Overall Recommendation:** 3

**Summary:**

Interesting topic and good experimental design choices. However the work lacks empirical evidence that watermarking is what makes the method strong.

**Claims And Evidence:**

Good: LLM rephrasing enables reliable statistical tests for dataset membership.

Bad: The fact that it works better with green/red watermarking is not proven. It seems like adding a distortion (and so making the reformulation worse perplixity-wise) increases memorization and thus the strength of the test. However, the authors do not test simply rephrasing with higher temperature and without watermarking, which could be a simpler solution. If it works, watermarking is not necessary.

**Essential References Not Discussed:**

It lacks related works to radioactivity of watermarks see https://arxiv.org/abs/2402.14904, i.e. how to find traces of watermarks in the model.  FYI the same authors have (apparently post ICML deadline) posted a paper on how to use that to protect benchmarks specifically, without the need for multiple reformulations

**Experimental Designs Or Analyses:**

yes, see "claims & evidence" section.

Main limitation is that the emphasis is mostly on protecting benchmarks, while the method makes more sense for other types of texts: if one is okay to keep some versions of the benchmark private, than it couldnt they keep the original benchmark private?

**Methods And Evaluation Criteria:**

Yes, the benchmarks and models evaluated make sense. As stated as a limitation however, it would have been better to have some pretraining experiments. Within the compute budget used in this work, the  authors could have made them.

The main limitation that I see is to keep x copies of a benchmark, they could also leak. Also, you could just keep your benchmark private the whole time. That way you are sure that no LLM is contaminated! For protection of other types of texts, the authors' methods make more sense though.

**Other Comments Or Suggestions:**

see above

**Other Strengths And Weaknesses:**

- Very nicely written
- Important line of work, and the idea of rephrasing is nice in order to have good calibration and enable Dataset inference
- Good experimental design choices, except IMO too much emphasis on benchmark protection

**Questions For Authors:**

- discuss litterrature on radioactive watermarks
- add an experiments where instead of watermarking with green/red list during reformulation, just use a higher temperature. In other words, add a scatter plot with p-value (y-axis) and perplexity of the reformulation (x-axis), with or without watermarking but different temperature. Perplexity is probably the key factor here
- Again about perplexity, what makes the method works is the distortion. For benchmarks, the authors show that its okay because it keeps model's accuracy. However, for other texts, people would maybe not want to publish online a reformulated version of their text with flaws. I sense that the authors want to pass the message that there is no flaws most of the time: the previous experiment (p-value as a function of perplexity) would highlight this point. Otherwise, this should be more clearly stated as a limitation.

**Relation To Broader Scientific Literature:**

It compares to two good baselines, and the related work is pretty complete

**Theoretical Claims:**

No theoretical claim.

---

> ### Author Rebuttal · Authors · 2025-04-01
>
> We thank the reviewer for their positive assessment of our work. We respond to the raised concerns below.
>
> ### Re: Experimental Designs
>
> > E1: Main limitation is that the emphasis is mostly on protecting benchmarks, while the method makes more sense for other types of texts
>
> We would like to clarify that our study is not limited to  protecting benchmarks. Our focus is on the broader problem of dataset inference. We conduct preliminary experiments on detecting benchmark contamination due to its importance  and the strict requirements for ensuring quality of the rephrased versions. Importantly, in Section 5, we demonstrate STAMP can successfully detect membership of diverse real-world content **including blog articles and research abstracts.**
>
> > E2: if one is okay to keep some versions of the benchmark private, than it couldnt they keep the original benchmark private?
>
> We agree with the reviewer that private benchmarking presents one solution to the contamination problem. However, as highlighted by previous work [1], private benchmarking raises concerns about transparency. We believe our work contributes to an ongoing discussion on ensuring trustworhty evaluations and provides a method to detect contamination for public benchmarks.
>
> ### Essential References
>
> > R1: Radioactive Watermarks (also Q1)
>
> In [2], authors demonstrate that it's possible to detect if an LLM is **fine-tuned** on the outputs of another watermarked LLM, by detecting the watermark signal in the outputs. A follow-up work [3] (posted the ICML deadline) extends this framework to detect benchmark contamination through watermarked rephrasing, similar to our approach. We find that their approach has limited applicability since it requires the rephrasing LLM and the contaminated LLM to share the same tokenizer. Moreover, we hypothesise that their approach would require stronger watermarks and higher repetition in training data. We conduct preliminary experiments to verify our hypothesis.
>
> Our results show that while [3] can detect contamination with higher repetitions and stronger watermarks, STAMP significantly outperforms this approach across all settings (lower p-value is better, with p < 0.05 indicating contamination and ~0 denotes vanishingly small p values).
>
> | Watermark Strength | Repetition | P value |  |
> |---|---|---|---|
> |  |  | STAMP | [3] |
> | 2.0 | 1 | 6.8e-28 | 0.65 |
> | 2.0 | 4 | ~0 | 1.1e-01 |
> | 4.0 | 4 | ~0 | 4.7e-3 |
>
> ### Sampling with a higher temperature
>
> We thank the reviewer for suggesting an interesting experiment (Q2). We use this section to discuss reviewer's concern about whether watermarking is necessary and instead if we could just use a higher temperature.
>
> We would like to note that while one intuition behind our approach to use watermarking is indeed increasing memorisation (due to distortions as pointed by the reviewer), we also highlight another important intuition: using different keys (as hash keys) for different rephrases embeds distinct watermarking signals, **reducing token overlap between versions** and increasing perplexity divergence specifically in cases of contamination (since the contaminated model would overfit the tokens in the public rephrase).
>
> Our position is that while sampling with a high temperature is important, the use of green/red list watermarking is complementary and helps in improving the sensitivity of our test. We perform some preliminary experiments by sampling with a temperature of $1.2$ and found our initial results to be negative. We compute the std dev. of ppl across rephrasing of  each sample and present the 95 percentile value below. Our results show that at higher T ($T$ > 1.2) the model often generates gibberish, which renders our test ineffective to the large and frequent outliers. While we believe in principle, we could calibrate the T better, our initial results show that increasing T beyond a point might not be optimal and watermarking can be complementary to higher temperatures.
>
> | temp | Watermark Strength | 95 percentile std(ppl) |
> |---|---|---|
> | 1 | 0.0 | 40 |
> | 1 | 2.0 | 48 |
> | 1.2 | 0.0 | 3863 |
>
> ### Questions:
>
> Q1 and Q2 are addressed above.
>
> > Q3: Flaws due to the distortion
>
> This is indeed a valid concern and to address this concern, we performed a human study in Section 5 where we indeed find that 6 out of 24 authors indicated their abstracts could use **minor edits**, suggesting the rephrases can have flaws. However, **majority of the authors found the rephrasing to be acceptable.** We believe that this will be less of a concern going forward, as general model capabilities (including paraphrasing) continues to improve. Regardless, we will note this as a limitation in our draft.
>
> ---
>
> [1] Bansel et al. Peeking Behind Closed Doors: Risks of LLM Evaluation by Private Data Curators. ICLR 25 blog post
> [2] Sander et al. Watermarking Makes Language Models Radioactive. NeurIPS 24
> [3] Sander et al. Detecting Benchmark Contamination Through Watermarking. Preprint

---

### Official Review · Reviewer_7i4M · 2025-03-11

**Overall Recommendation:** 5

**Summary:**

The paper proposes a framework, called Stamp, for detecting dataset membership (infering whether a dataset was included in the pretraining dataset of an LLM).

The framework consists of generating multiple watermarked rephrases of the content, with a distinct watermark embedded in each rephrasing. One version is released publicly, the others kept private. When a model is then released, they compute the model perplexity on both the public version and the private versions. Using a  statistical test to compare model likelihoods, they can make an informed inference.

They specifically use the KGW watermarking scheme, steering generations towards a green subset of the vocabulary.

They test their approach by further pretraining pythia 1B on contaminated training data. They find their method to work very well, better than other approaches and able to identify even small contamination.

**Claims And Evidence:**

Yes. The only unfortunate thing is the connection to real-world pretraining data sizes.

**Essential References Not Discussed:**

I would not per se say essential, but the following pieces of work could contribute to a potentially better positioning of the work:

1. A paper showing that it is feasible to detect whether models are trained on watermarked content, could be a justification in why watermarking is chosen as a technology to do this. Sander, T., Fernandez, P., Durmus, A., Douze, M., & Furon, T. (2024). Watermarking makes language models radioactive. Advances in Neural Information Processing Systems, 37, 21079-21113.

2. The benchmark big-bench actually includes a 'canary' in their benchmark which could enable the detection as well.I don't think it's particularly effective compared to STAMP, but it does feel like something relevant in the related work.  Srivastava, A., Rastogi, A., Rao, A., Shoeb, A. A. M., Abid, A., Fisch, A., ... & Wang, G. (2022). Beyond the imitation game: Quantifying and extrapolating the capabilities of language models. arXiv preprint arXiv:2206.04615.

3. A position paper arguing that MIAs cannot prove that an LLM was trained on certain data. Zhang, J., Das, D., Kamath, G., & Tramèr, F. (2024). Membership inference attacks cannot prove that a model was trained on your data. arXiv preprint arXiv:2409.19798.

**Experimental Designs Or Analyses:**

Yes. Their main experiment on training pythia 1B on a deliberately contaminated dataset makes a lot of sense. All the ablations are well thought through and add valuable insights.

**Methods And Evaluation Criteria:**

Yes.

**Other Comments Or Suggestions:**

- Another limitation of methods used by Wei et al. and Meeus et al. is the possibility that these unique sequences or copyright traps are removed by training data preprocessing techniques (either by perplexity filtering or by deduplication) - which stamp is not prone to. It might be good to also mention that as a justification.

- Currently you have two different citations of Meeus et al. (copyright traps for LLMs), and it's not clear whether this is referring to two different papers or consistently to the same.

**Other Strengths And Weaknesses:**

Strengths:
- The paper is very well written, the method is clearly novel and works well. Overall a great piece.
- The analysis that a bag of words classifier can distinguish between human generated text and LLM-based rephrases is interesting and very useful. Authors might find it to useful to relate this to similar mistakes made in the field of membership inference attacks [1,2].
- The paper includes the important ablation studies that I as a reviewer thought about when reading, and executes them very carefully. Specifically: (i) The use of watermarked rephrases instead of regular rephrases, (ii) Maintaining the utility of the benchmark after the watermarked rephrases and (iii) When only a part of the benchmark is used for training.

Weaknesses:
- I find the experimental setup to be quite convincing, but the entire analysis does not compellingly show if this would scale to real-world pretrained models. Figure 3 shows that with pretraining data size, the method becomes less effective, while the scale of 7B tokens does not come close to the scale of today's pretraining datasets.  However, they still show that this is better than any other method available today in their setup, so this is probably ok.
- The paper could improve if they consider different watermarking schemes.


[1] Das, D., Zhang, J., & Tramèr, F. (2024). Blind baselines beat membership inference attacks for foundation models. arXiv preprint arXiv:2406.16201.

[2] Meeus, M., Shilov, I., Jain, S., Faysse, M., Rei, M., & de Montjoye, Y. A. (2024). SoK: Membership Inference Attacks on LLMs are Rushing Nowhere (and How to Fix It). arXiv preprint arXiv:2406.17975.

**Questions For Authors:**

- I appreciate the ablation done for stamp using rephrases that are watermarked versus not watermarked. From table 1, it seems that the not watermarked versions using stamp also work well. Are there any trade-offs to be made to use the watermarked version?

- For the scale of real-world models and benchmarks, what is the value of private key count you would actually recommend? And how many data samples would you need for it to be meaningful? Could you apply any scaling laws to results such as in Figure 3?

**Relation To Broader Scientific Literature:**

They consider very relevant pieces of work to position themselves.

- They convincingly show that Zhang et al.'s approach suffers from a distribution shift, making the contamination detection flawed in practice.
- They cite both Wei et al. and Meeus et al., who use unique sequences to mark content. They argue that these techniques impair machine readability, indexing and retrieval- making it impractical for content creators. For benchmarks, they argue that these techniques might alter their utility. These arguments are fairly convincing.

**Theoretical Claims:**

NA

---

> ### Author Rebuttal · Authors · 2025-04-01
>
> We thank the reviewer for their insightful comments and are happy to see that the reviewer enjoyed our writing and found our method novel. We respond to the reviewer’s comments and questions below.
>
> ### Re: Weaknesses
>
> > W1: Scaling to real-world pretrained models.
>
> We acknowledge the reviewer's concern about our detector scaling to real-world models. However as the reviewer points out, our method is better than any other method available today. We believe that with larger LLMs and some amount of repetition (factors known to increase memorization [3]) our method will be effective at the scale of real-world datasets and will, hopefully, show better scaling properties compared to other approach. We would like to refer the reviewer to our response to Reviewer ojw8, where we show that for the same $\frac{token}{parameter}$, larger models yield stronger detection results.
>
> ### Re: Questions
>
> > Q1: I appreciate the ablation done for stamp... Are there any trade-offs to be made to use the watermarked version?
>
> We agree that the version of STAMP without watermarked variants also works well, but we demonstrate that watermarking increases the statistical strength of our test. This suggests better scaling trends with larger pretraining datasets. In principle, using a stronger watermark signal may increase detectability but may hurt the quality and utility of the paraphrased text.
>
> > Q2: For the scale of real-world models and benchmarks
>
> >> (2a): what is the value of private key count you would actually recommend?
>
> We believe our analysis in Figure 2 (right) is largely independent of scale of model and benchmarks. We find that 5-6 keys should suffice, as at this point the empirical average of each test sample closely approximates the true average, a factor we believe is independent of the scale of model and benchmarks.
>
> >> (2b): And how many data samples would you need for it to be meaningful?
>
> Our test provides statistically significant results with as few as a few hundred test pairs (500-1000). Importantly, as noted in our response to Reviewer aozb, our framework allows a single copyrighted sample to generate multiple test pairs.
>
> >> (2c): Could you apply any scaling laws to results such as in Figure 3?
>
> Thanks for the suggestion. We conduct additional analyses to apply scaling laws to our results. Using the data points from Figure 3, we fit the power law from [3] (linear relationship between $log\( \text{p value} \)$ and $log(D)$). We obtain a good fit (e.g. $r^2$ (goodness of fit) of 0.9 for trivia_qa) with the resulting curves predicting that our method will obtain statistically significant results (p<0.05) upto $\approx10$B tokens.
>
> ### Re: Essential References:
>
> We agree with the reviewer that while these missing references are not essential, discussing them would help us better position our work.
>
> > R1: Watermarking makes language models radioactive
>
> We refer the reviewer to our rebuttal to Reviewer oVrP for detailed discussion on radioactive watermarks including additional experiments. We will include these in the next version of the our draft.
>
> > R2: `Canary` in the big-bench  benchmark
>
> We believe that while these canaries were originally designed for a different purpose, in principle, they could serve a similar detection function as Wei et al.'s random sequence insertion method. Though potentially effective in some scenarios, such an approach would face the same limitations we highlight for Wei et al.'s work.
>
> > R3: Membership inference attacks cannot prove that a model was trained on your data.
>
> The position paper [3] argues that existing MIAs that use data collected a posteriori for calibration are statistically unsound, a position that aligns with our analysis Zhang et al.'s approach, where we highlight it suffers from a distribution shift. More importantly, we believe our framework meets the criteria of a sound training data proof argued in the position paper. Our test rejects the null hypothesis when the test statistic on the public dataset is unusual compared to private datasets which a priori were equally likely to have been used for training.
>
> > C2: Two different citations of Meeus et al. (copyright traps for LLMs).
>
> Thanks for pointing this out! These refer to the same paper and we will fix it in the draft.
>
> ---
>
> Once again, we thank you for the constructive feedback. Working on the questions has helped us improve the quality of our analysis. Please let us know if we can address any further concerns.
>
> [3] Kaplan et al. Scaling Laws for Neural Language Models
> [4] Zhang et al. Membership inference attacks cannot prove that a model was trained on your data. SaTML 25
> [5] Carlini et al. Quantifying Memorization Across Neural Language Models. ICLR 23

---

### Official Review · Reviewer_aozb · 2025-03-14

**Overall Recommendation:** 3

**Summary:**

The authors propose a method for dataset membership inference based on generating one public paraphrase of specific content and several private ones, then using a perplexity-based statistical test for detecting whether the dataset was part of the training set.

### Update after rebuttal: Thank you for addressing my concerns. I updated my score

**Claims And Evidence:**

The claims appear to be supported by enough evidence.

**Essential References Not Discussed:**

N/A

**Experimental Designs Or Analyses:**

One potential issue is that the authors consider only continual pretraining. The concern is that the samples seen toward the end of training are more likely to be "fresher in the LLM's memory," so the results for regular pretraining (which is a more realistic scenario) might differ significantly.

**Methods And Evaluation Criteria:**

The proposed method and evaluation criteria seem sound to me.

**Other Comments Or Suggestions:**

N/A

**Other Strengths And Weaknesses:**

I noticed a few weaknesses that I hope the authors can clarify or address. If the goal is to detect copyrighted samples, it seems like a strong assumption to presume that the "defender" has a relatively large dataset of copyrighted samples. It could be the case that they only have a few samples, which might render the method ineffective. On the other hand, if the goal is to detect benchmarks in the model's pretraining, I don't think it's realistic to assume that someone could know in advance that a benchmark might be used for a model, then ensure it is paraphrased everywhere it appears on the internet, so they can rely on dataset membership detection later. Additionally, if all the benchmarks from the model's pretraining were "paraphrased," this could negatively affect performance, as the model might start learning the "watermarks" introduced by the paraphrases.

**Questions For Authors:**

N/A

**Relation To Broader Scientific Literature:**

The proposed method improves over the considered baselines. However, I am not fully familiar with the dataset membership inference literature.

**Theoretical Claims:**

N/A

---

> ### Author Rebuttal · Authors · 2025-04-01
>
> We thank the reviewer for their feedback. We respond to the reviewer’s comments and questions below.
>
> ### Re: Weaknesses
>
> > W1: If the goal is to detect copyrighted samples, it seems like a strong assumption to presume that the "defender" has a relatively large dataset of copyrighted samples. It could be the case that they only have a few samples, which might render the method ineffective.
>
> We would like to clarify a potential misunderstanding about the sample complexity of our detection method. We find that our test yields statistically significant results with as few as **400 pairs** (Figure 3). Importantly, a single copyrighted sample can generate multiple test pairs. For instance, in our blog post case study (section 5.2), we perform dataset inference using **44 posts** by performing dataset inference on a collection of paragraphs from the blog, each paragraph forming a separate test pair. This approach allows our method to work effectively even with limited copyrighted samples, giving content creators flexibility in defining what constitutes a pair for the statistical test.
>
> > W2: On the other hand, if the goal is to detect benchmarks in the model's pretraining, I don't think it's realistic to assume that someone could know in advance that a benchmark might be used for a model, then ensure it is paraphrased everywhere it appears on the internet, so they can rely on dataset membership detection later.
>
> We would like to clarify that our approach is not meant to protect existing benchmarks but we offer a solution for future dataset releases. As per our approach, the **onus is on the benchmark creators to watermarks their own benchmarks before releasing them online.** The creator generates multiple paraphrases, each with a unique watermarked key. One version is released publicly for model evaluation, while others remain private, as explained in section 4.1 of our draft. To detect contamination, benchmark creators can apply our statistical test **on any target model** to obtain evidence if the **target model's** training data included their benchmark. We highlight this requirement in Section 7 as a limitation, noting that **data must be carefully prepared before public release** to enable detection. This constraint is not unique to our approach but is shared by existing works [1,2] and we believe is fundamental for a sound statistical test [3].
>
> > W3: Additionally, if all the benchmarks from the model's pretraining were "paraphrased" this could negatively affect performance, as the model might start learning the "watermarks" introduced by the paraphrases.
>
> We would like to point out that an important feature of our work is that it allows benchmark creators to _use different private hash keys to watermark the public version of each document in their collection_. Given that each dataset only constitutes a small fraction of overall training corpora, and with prior work [4] demonstrating that even training on data with just two different keys doesn't result in models learning individual watermarks, it is unlikely that the model would learn the watermark. Hopefully, this assuages your concern about potential watermarking learning.
>
> Additionally, we experiment (in the section 4.2; under false positive analysis) to detect membership of held out samples that are watermarked with the same key as the contaminated samples, and find that our approach (correctly) does not detect them to a part of the training corpus. **This experiment provides further evidence that models do not learn the watermarks introduced by the paraphrasers.**
>
> ### Re: Experimental Design
>
> > One potential issue is that the authors consider only continual pretraining... results for regular pretraining... might differ significantly.
>
> We acknowledge the reviewer's concern about the potential recency bias in our setup. While previous research [5] has shown that training order has little effect on memorization, the training dynamics of memorization in LLMs remains an active area of research. While such factors can influence the memorization, our contribution is in demonstrating for any given level of memorization, our test provides greater sensitivity compared to any other existing approach and is robust against false positives.
>
> ---
>
> Once again, we thank you for the insightful questions. We hope this response has addressed your concerns and would request you to kindly reconsider your overall assessment. Please let us know if we can address any further concerns.
>
> [1] Wei et al. Proving membership in LLM pretraining data via data watermarks. ACL 24
> [2] Oren et al. Proving Test Set Contamination in Black-Box Language Models. ICLR 24
> [3] Zhang et al. Membership Inference Attacks Cannot Prove that a Model Was Trained On Your Data. SaTML 25
> [4] Gu et al. On the Learnability of Watermarks for Language Models. ICLR 24
> [5] Biderman et al. Pythia: A Suite for Analyzing Large Language Models Across Training and Scaling. ICML 2023

---

### Official Review · Reviewer_ojw8 · 2025-03-27

**Overall Recommendation:** 2

**Summary:**

This paper presents STAMP, a framework that helps content creators detect whether their content (e.g., benchmark test sets, blog articles, research abstracts) has been used without authorization in the pretraining of large language models. The key idea is to release watermarked rephrasings of the content, which embed subtle signals via a known watermarking scheme. By generating multiple watermarked variants for each piece of content (one “public” version is released online, the rest are kept private), one can perform a paired statistical test on the target model’s perplexities. If a model’s likelihood of producing the “public” (known) watermark version is systematically higher than for private/unreleased variants, it strongly indicates that the model was trained on that public data. Crucially, the authors show that this approach outperforms prior membership-inference or contamination-detection methods, even when each dataset appears only once (at a minuscule proportion of the training tokens). They also confirm that STAMP’s watermarked versions do not degrade the usability (accuracy or ranking) of the benchmark or text.

**Claims And Evidence:**

1. Claim: The authors can reliably detect dataset membership for text that was used once in a massive training set.

Evidence: In controlled experiments on 1B-parameter “Pythia” LLM variants, with four benchmarks contaminated at <0.001% of tokens, STAMP still achieves low p-values (e.g. 10^-4 to 10^-6).

2. Claim: Watermarking-based rephrasings enhance memorization signals, thereby boosting detection sensitivity.

Evidence: They compare watermarked vs. non-watermarked rephrasings and show that the watermarked versions yield significantly stronger results (two orders of magnitude in p-values).

3. Claim: STAMP preserves the “utility” of a dataset (e.g., a benchmark’s function as a measure of model performance).

Evidence: They evaluate standard LLMs on both the original and watermarked benchmark copies. The absolute accuracy remains nearly the same, and crucially, the relative ranking of models is unchanged, demonstrating that the test’s transformations do not distort the difficulty or nature of the underlying tasks.

4. Claim: STAMP effectively avoids false positives.

Evidence: When applying STAMP to models that never saw the watermarked dataset, the resulting p-values show no spurious membership detection. Similarly, watermarked hold-out sets from the same domain also do not register as “in the training data,” indicating the method is capturing genuine membership.

**Essential References Not Discussed:**

1. Important baselines not discussed or compared: there has been a lot of membership inference methods for contamination detection [1,2,3]. The authors only focused on the curated experiments in the paper but there are other available benchmarks in these papers or the authors did not evaluate these methods on the proposed data and models in this paper.

2. There are other data contamination papers that conducted similar experiments in the paper, though from different perspectives but the methodology are quite similar [4,5].

Many recent literatures are missing, not limited to what mentioned above.

[1] Shi, Weijia, et al. "Detecting pretraining data from large language models." arXiv preprint arXiv:2310.16789 (2023).
[2] Zhang, Jingyang, et al. "Min-k%++: Improved baseline for detecting pre-training data from large language models." arXiv preprint arXiv:2404.02936 (2024).
[3] Zhang, Weichao, et al. "Pretraining data detection for large language models: A divergence-based calibration method." arXiv preprint arXiv:2409.14781 (2024).
[4] Jiang, Minhao, et al. "Investigating data contamination for pre-training language models." arXiv preprint arXiv:2401.06059 (2024).
[5] Yang, Shuo, et al. "Rethinking benchmark and contamination for language models with rephrased samples." arXiv preprint arXiv:2311.04850 (2023).

**Experimental Designs Or Analyses:**

The authors mainly conduct two main sets of experiments:
1. Benchmarks: Inject 4 standard test sets (TriviaQA, ARC, MMLU, GSM8K) once into a ~6.7B token corpus for a 1B-parameter model (Pythia-1B). Even though each dataset is <0.001% of the training data, STAMP reliably detects contamination (p<1e-4 to 1e-6).
2. Case Studies: Paper abstracts (EMNLP 2024) and AI Snake Oil blog posts. They show that STAMP can also detect these real-world sets in the model’s training data.

**Methods And Evaluation Criteria:**

Yes, the authors evaluated the effectivenss of STAMP by continuing pre-training the Pythia-1B on deliberately contaminated pretraining data.  The evaluation process includes p-values from the paired t-test measure how convincingly the model “prefers” the public watermarked text; AUROC for membership inference attacks (used as comparative baselines); Effect on Utility: They check if watermarked datasets still produce valid measures of model performance (preserving the rank-order and approximate accuracy).

**Other Comments Or Suggestions:**

1. It might be helpful to test the method on even larger, more highly capable LLMs. The paper suggests it should generalize well, since bigger models memorize more.
2. How does the proposed method deal with the rephrased or intended contamination as mentioned in [1,2]?

[1] Jiang, Minhao, et al. "Investigating data contamination for pre-training language models." arXiv preprint arXiv:2401.06059 (2024).
[2] Yang, Shuo, et al. "Rethinking benchmark and contamination for language models with rephrased samples." arXiv preprint arXiv:2311.04850 (2023).

**Other Strengths And Weaknesses:**

Strengths:

1. The paper is easy to follow.
2. Applicable to real text of various lengths (from short question sets to multi-paragraph blog posts).
3. Achieves robust detection even with only a single copy of each example in the training set.

Weaknessses:
1. Relies on “grey-box” access, meaning you can query token probabilities from the suspect model (some commercial APIs may not allow direct logit or perplexity queries). And as mentioned above, the experiments did not include the comparison with other white-box or grey-box methods, whic pose real applicability of the proposed method.

**Questions For Authors:**

See above

**Relation To Broader Scientific Literature:**

Data Contamination is an important problem that ties into recent concerns of test-set contamination (particularly for LLMs) and how one can detect or mitigate it. There has been a lot of recent studies that focus on this direction.

**Theoretical Claims:**

There are no proposed new theoretical proofs for this paper. The paper mainly focuses on application and experimental results.

---

> ### Author Rebuttal · Authors · 2025-04-01
>
> We thank the reviewer for their insightful comments and feedback. We are happy to see that they appreciate the robust detectability that our method offers, and find the paper easy to follow. We discuss their concerns below:
>
> ### Re: Important baselines not discussed (ER1)
>
> Thanks for sharing these baselines, we would like to clarify that in our current submission, we already compare our approach against *min-k* [1] (along with other popular MIAs) in section 4.2 under baselines and Table 7, and find our approach to outperform these baselines.
>
> Based on your suggestions, we conduct new experiments to benchmark additional MIAs [2,3] and share the detection performance (AUROC) under two settings of non-members:`same documents` where we use different rephrasing of the same test samples and `different documents` where we use a held out set of test samples. The AUROC scores $\approx 0.5$ show that such baselines are ineffective in determining membership.
>
> | Dataset | Same Documents |  | Different Documents |  |
> |---|---|---|---|---|
> |  | **Min-k++** [2] | **DC-PDD** [3] | **Min-k++** [2] | **DC-PDD** [3] |
> | TriviaQA | 0.50 | 0.52 | 0.44 | 0.58 |
> | ARC-C | 0.49 | 0.51 | 0.45 | 0.52 |
> | MMLU | 0.49 | 0.52 | 0.45 | 0.52 |
> | GSM8k | 0.50 | 0.52 | 0.48 | 0.52 |
>
> Overall our findings corroborate with recent studies [6,7] that highlight the failure of such heuristic based MIAs. Additionally, as highlighted by previus work [8], such heuristic MIAs do not provide a sound statistical proof of membership. Given these additional experiments, we hope that your major concern is addressed.
>
> ### Re: Other data contamination papers (ER2)
>
> We respectfully disagree with the assertion that our methodology is "quite similar" to the cited works. [5] primarily studies how inclusion of simple variations of test data (including rephrasings) can artificially inflate benchmark performance. Similarly, [4,5] highlight limitations of existing white-box decontamination approaches. In contrast, our approach utilizes _watermarked rephrasings_ as a component of our proposed framework which provides a principled statistical framework for dataset inference. We will update the draft to discuss and constrast our work from these related papers [4,5].
>
> ### Re: Weakness: “grey-box” access
>
> We acknowledge your concern about our work requiring grey-box access to model probabilities (also discussed as a limitation in Section 7 of our draft). However, majority of prior work on dataset inference assumes grey-box access, similar to our work. In the current landscape where there is a lack of successful _black-box MIAs that provide a statistical proof_, alternative approaches can be used: probability extraction attacks [9] or through trusted third parties such as legal arbiters. Importantly, as the field develops better black-box metrics, our statistical framework may be adapted to perform paired tests on those metrics instead of loss values.
>
> ###  Re: Other Comments
>
> > It might be helpful to test the method on even larger, more highly capable LLMs. The paper suggests it should generalize well, since bigger models memorize more.
>
> We agree that testing on larger LLMs would be valuable. Existing research [10] shows that larger models memorize training data more aggressively. We present preliminary results comparing models of different sizes, maintaining a proportional token-to-parameter ratio:
>
> | parameter ($P$) | token ($T$) | ARC-C | GSM8K | TriviaQA | MMLU  |
> |----------|-------|-------|-------|----------|-------|
> | 410m     | 400m       | -16.5 | -28.5 | -16.7    | -8.4  |
> | 1000m    | 1000m      | -21.7 | -39.8 | -27.8    | -16.8 |
>
> The results ($log \( \text{p value} \)$ , where lower is better and anything below -3 is statistically significant) suggest that for a fixed $\frac{T}{P}$, our method performs better on larger models, supporting our hypothesis about improved detection with model scale.
>
> > Dealing with the rephrased or intended contamination?
>
> Thanks for the insightful question! To clarify, our work focuses specifically on verbatim contamination, as our primary goal is to detect membership of a dataset with statistical guarantees. We will update Section 7 to explicitly discuss this point, and clarify the scope of our study.
>
> ---
>
> Once again, we thank the reviewer for the constructive feedback. We hope our response addresses their concerns effectively and kindly request you to reconsider your assessment of our work. Please let us know if we can address any other concerns.
>
> [6] Duan et al. Do membership inference attacks work on large language models? COLM 24
> [7] Maini et al. LLM Dataset Inference: Did you train on my dataset? NeurIPS 24
> [8] Zhang et al. Membership Inference Attacks Cannot Prove that a Model Was Trained On Your Data. SaTML 25
> [9] Morris et al. Language Model Inversion. ICLR 24
> [10] Carlini et al. Quantifying memorization across neural language models. ICLR 23

---

### Decision · Program_Chairs · 2025-05-01

**Decision:**

Accept (poster)

**Comment:**

This paper presents STAMP, a method for detecting dataset contamination in large language models using statistical tests over watermarked paraphrases. The proposed framework allows dataset owners to release a public, watermarked version of their content and retain private paraphrases. The paper solves an important problem with a well-designed methodology and careful evaluation. While two reviewers did not engage post-rebuttal, their concerns were substantively addressed with new experiments and clarifications.